# Transolver-3: Scaling Up Transformer Solvers to Industrial-Scale Geometries

**Hang Zhou** [1]  **Haixu Wu** [1]  **Haonan Shangguan** [1]  **Yuezhou Ma** [1]  **Huikun Weng** [1]  **Jianmin Wang** [1]
**Mingsheng Long** [1]

## Abstract

Deep learning has emerged as a transformative tool for the neural surrogate modeling of partial differential equations (PDEs), known as neural PDE solvers. However, scaling these solvers to industrial-scale geometries with over $10^8$ cells remains a fundamental challenge due to the prohibitive memory complexity of processing high-resolution meshes. We present Transolver-3, a new member of the Transolver family as a highly scalable framework designed for high-fidelity physics simulations. To bridge the gap between limited GPU capacity and the resolution requirements of complex engineering tasks, we introduce two key architectural optimizations: faster slice and deslice by exploiting matrix multiplication associative property and geometry slice tiling to partition the computation of physical states. Combined with an amortized training strategy by learning on random subsets of original high-resolution meshes and a physical state caching technique during inference, Transolver-3 enables high-fidelity field prediction on industrial-scale meshes. Extensive experiments demonstrate that Transolver-3 can handle meshes with over 160 million cells, achieving impressive performance across three challenging simulation benchmarks, including aircraft and automotive design tasks. Code is available at https://github.com/thuml/Transolver-3.

## 1. Introduction

Partial differential equations (PDEs) are fundamental to major scientific and engineering problems (Roubíček, 2005). Since analytic solutions for complex PDEs are usually hard to obtain, numerical methods are widely used, such as in weather forecasting (Bauer et al., 2015) and automotive

[1]School of Software, BNRist, Tsinghua University, China. Hang Zhou<zhou-h23@mails.tsinghua.edu.cn>. Correspondence to: Mingsheng Long <mingsheng@tsinghua.edu.cn>.

*Proceedings of the 43rd International Conference on Machine Learning*, Seoul, South Korea. PMLR 306, 2026. Copyright 2026 by the author(s).

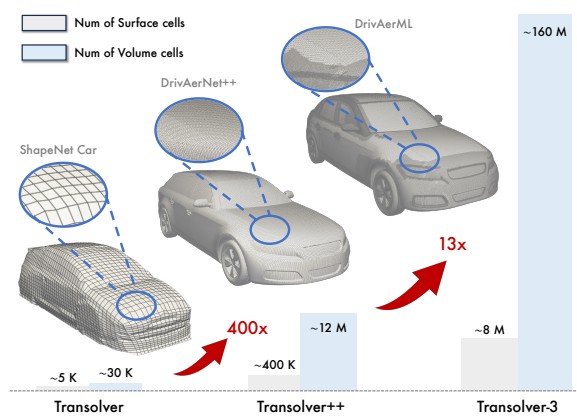

*Figure 1.* The maximum mesh sizes handled by Transolver series.

aerodynamics (Tu et al., 2023). However, high-fidelity simulations remain computationally expensive, often requiring hours or even days to obtain a single solution (Ashton et al., 2024b). To this end, neural networks have recently emerged as promising surrogate models, known as neural PDE solvers (Li et al., 2021), to accelerate the solving process. The powerful non-linear approximation capability of neural networks enables them to directly learn mappings from input parameters to solutions. Once trained, these surrogate models can offer the potential for order-of-magnitude accelerations in industrial design workflows. This potential has been recognized by the industry, leading to the integration of neural surrogates into Computer-Aided Engineering (CAE) softwares, such as Ansys SimAI (Ansys, Inc., 2024) and Altair PhysicsAI (Altair Engineering Inc., 2024).

Despite the progress of neural PDE solvers, real-word industrial applications often involve large-scale meshes, the size of which can easily exceed 100 million (Ashton et al., 2024b). The necessity of such massive meshes is driven by both physics fidelity and numerical stability. For example, in industrial Computational Fluid Dynamics (CFD), transitioning from Reynolds Averaged Navier-Stokes (RANS) simulations to scale-resolving simulations like Large Eddy Simulations (LES) offers increased fidelity of physical phenomena like large-scale flow separation, but comes at the cost of increased mesh resolution and computational overhead (Spalart, 2000). Processing massive-scale geometries poses great challenges for neural PDE solvers. Specifically, a single intermediate activation tensor for a 100-million

cell mesh with a feature dimension of 128 requires around 51.2 GB of memory in FP32 precision. In a common multi-layer neural solver, the cumulative memory footprint quickly exceeds the 80 GB capacity of high-end GPUs. Such huge resource requirement is computationally and economically prohibitive for most practical scenarios. Thus, *geometry scaling*, the capability of scaling up to industrial-scale geometries under current hardware limitations is the key to a practical neural PDE solver.

As a representative work of recent progress in neural PDE solvers, Transolver (Wu et al., 2024b) excels at capturing intricate physical correlations on complex geometries. Its primary strength lies in the Physics-Attention mechanism, which offers both high prediction accuracy and good interpretability by aggregating discrete mesh information into intrinsic physical states. The geometry flexibility makes Transolver an ideal candidate for scaling toward real-world engineering problems. However, Transolver also faces significant memory bottlenecks, with the maximum input size limited to 700k cells (Luo et al., 2025). While Transolver++ (Luo et al., 2025) tries to mitigate this by leveraging multi-GPU parallelism to handle million-scale meshes, extending this approach to industrial-scale problems with over $10^8$ mesh cells would require hundreds of GPUs (Alkin et al., 2025). To address these challenges, we propose Transolver-3, an optimized framework for industrial-scale geometries. Our approach systematically resolves scalability bottlenecks both at the training and inference phases:

*(i) During training*, the linear scaling of intermediate memory footprint with respect to mesh size renders high-resolution training intractable for industrial geometries.

*(ii) During inference*, the goal of high-fidelity prediction necessitates information aggregation from the full mesh.

During the training phase, we conduct a detailed complexity analysis of the original Physics-Attention mechanism, discover and implement two structural optimizations to maximize single-GPU throughput. While these refinements enable Transolver-3 to achieve approximately $1.9\times$ single-GPU capacity than Transolver++, processing industrial-scale geometries exceeding $10^8$ cells remains hardware-prohibitive. To bridge this gap, we introduce a geometry amortized training strategy, allowing the model to learn the underlying physics laws by training on random subsets of the high-fidelity mesh. For the inference phase, we decouple the physical state estimation from field prediction. By aggregating global information into a physical state cache, Transolver-3 achieves high-fidelity predictions on the full mesh. This ensures that Transolver-3 captures the full information of large-scale meshes without exceeding memory limits. Our contributions are summarized as follows:

- We propose architectural optimizations to the Physics-Attention mechanism and a geometry amortized training strategy, which collectively bridge the gap between hardware constraints and industrial-scale geometries.

- We introduce a decoupled inference framework that separates physical state estimation from field prediction. By aggregating global information into physical state caches, Transolver-3 enables high-fidelity predictions on full-resolution meshes.

- Transolver-3 achieves consistent state-of-the-art performances across three challenging industrial-level benchmarks with hundred-of-million scale meshes, highlighting the model's impressive efficiency and scalability.

## 2. Related Work

### 2.1. Neural PDE Solvers

Neural PDE solvers have emerged as promising alternatives of numerical solvers to accelerate the solving process. Existing neural PDE solvers can be roughly categorized into two paradigms. The first paradigm is the physics-informed neural networks (PINNs) (Raissi et al., 2019), which formalize PDEs as loss functions, yet they often suffer from optimization challenges and limited generalization (Wang et al., 2022; Wu et al., 2024a). The second is neural operators, which learn mappings between input functions and solutions from data. Among various methods, FNO approximates the kernel integral operator in the Fourier domain, while Geo-FNO (Li et al., 2023a) extends this to irregular geometries via domain deformation. LSM (Wu et al., 2023) enhance efficiency by compressing complex meshes into compact latent space. Graph Neural Networks (Scarselli et al., 2008) have also been introduced as they are well suited for irregular geometries. Representative methods include Graph-UNet (Gao & Ji, 2019), GNO (Li et al., 2020b), MeshGraphNet (Sanchez-Gonzalez et al., 2020) and GINO (Li et al., 2023b). MeshGraphNet performs message passing directly on simulation meshes to capture mesh-based interactions. GINO integrates GNO with Geo-FNO to capture both local and global correlation.

Transformers (Vaswani et al., 2017) have achieved remarkable success across many domains, and have also been explored for PDE solving. However, standard Transformer architectures suffer from quadratic computational complexity, limiting their scalability to large-scale meshes. Based on linear attention mechanism (Choromanski et al., 2021), models including OFormer (Li et al., 2023c), GNOT (Hao et al., 2023), and ONO (Xiao et al., 2024) avoid quadratic complexity. FactFormer (Li et al., 2023d) also introduces its efficient attention mechanism. To avoid applying attention directly over massive mesh points, methods such as Transolver (Wu et al., 2024b), UPT (Alkin et al., 2024), and GAOT (Wen et al., 2025) perform attention in the latent

space. Notably, Transolver introduces the Physics-Attention mechanism that groups mesh points into physical states and applies attention across these states for better efficiency.

Recent research has further advanced neural PDE solvers for generalization across multiple PDE families by scaling dataset diversity and parameter counts. Subramanian et al. investigate large-scale pretraining of FNO across three PDE families. MPP (McCabe et al., 2023) embeds diverse physics systems into a shared latent space. DPOT (Hao et al., 2024) conducts denoising auto-regressive pretraining with a Fourier Transformer. To further enhance generalizability, Unisolver (Zhou et al., 2025) integrates complete PDE components into pretraining, and Poseidon (Herde et al., 2024) employs time-conditioned architectures for flexible dynamics prediction. Despite their success in multi-PDE generalization, these models are mostly restricted to resolutions below $10^6$ cells, significantly lower than the $10^8$ cells possibly encountered in industrial-scale simulations.

## 2.2. Scaling to High-Resolution Geometries

Scaling neural PDE solvers to high-resolution meshes is essential for industrial applications. Early methods such as FNO and GNOT are largely limited to meshes on the order of $10^5$ cells. In contrast, latent-space architectures like UPT and Transolver have significantly extended scalability, with further advancements made by AB-UPT (Alkin et al., 2025) and Transolver++ (Luo et al., 2025), respectively. Specifically, AB-UPT achieves high-resolution outputs on industrial-scale meshes through multi-branch operators and anchored decoders, while Transolver++ leverages distributed parallelism to handle million-scale geometries. However, the scaling achieved by Transolver++ is primarily hardware-driven, which relies on multi-GPU parallelism without addressing the underlying algorithmic complexity.

The challenge of efficiently processing massive geometry data is similar to the long-context modeling challenges of Large Language Models (LLMs), where scaling to trillion-token corpora and long sequences has catalyzed critical breakthroughs in machine learning systems. For instance, Performer (Choromanski et al., 2021) exploits the associativity of matrix multiplication to reorder the attention computation, achieving linear time complexity and bypassing the storage of massive intermediate tensors. FlashAttention (Dao et al., 2022) significantly improves the efficiency of attention mechanisms through tiled computation and aggregation. For effective training of LLMs on long texts, sequence truncation paradigms (Vaswani et al., 2017) and their variants (Dai et al., 2019) are adopted. During inference, the KV cache, which avoids redundant recomputation of attention over historical tokens, serves as a critical technique for scaling LLM inference to long sequences. Inspired by these advances on long context handling in LLMs, Transolver-3 extends Transolver to handle industrial-scale geometries.

## 3. Transolver-3

Before introducing Transolver-3, we first revisit the Physics-Attention mechanism in Transolver and analyze its scalability challenges. After that, we demonstrate how Transolver-3 resolve the challenges at the training and inference phases.

### 3.1. Revisit Transolver Complexity

The Physics-Attention mechanism is the key of Transolver's geometry flexibility. As shown in Figure 2, given a mesh with $N$ cells represented as features $\mathbf{x} \in \mathbb{R}^{N \times C}$, the Physics-Attention mechanism begins by projecting $\mathbf{x}$ with two distinct linear layers. The first produces the projected latent feature $\mathbf{x}_{\text{proj}}$, the second generates slice weights $\mathbf{w} \in \mathbb{R}^{N \times M}$ with a following $\mathrm{Softmax}$ layer. Each $\mathbf{w}_{i,j}$ quantifies the contribution of the $i$-th mesh cell to the $j$-th physical state $\mathbf{s}_j$. The physical states are computed as a normalized weighted sum of $\mathbf{x}_{\text{proj}}$, named *slice* operation:

$$\begin{aligned} \mathbf{x}_{\text{proj}} &= \mathrm{Linear1}(\mathbf{x}) = \mathbf{x}\mathbf{w}_{\text{Linear1}}, \\ \mathbf{w} &= \mathrm{Softmax}(\mathrm{Linear2}(\mathbf{x})), \quad\quad (1) \\ \mathbf{s} &= (\mathbf{w}\mathbf{d}^{-1})^{\top}\mathbf{x}_{\text{proj}}. \end{aligned}$$

Here $\mathbf{d} \in \mathbb{R}^{M \times M}$ is a diagonal normalization matrix with $\mathbf{d}_{jj} = \sum_{i=1}^{N} \mathbf{w}_{ij}$. After self-attention, the updated states $\mathbf{s}'$ are *desliced* back to the mesh domain with the same slice weights, followed by a final projection layer:

$$\mathbf{x}_{\text{out}} = \mathrm{Linear3}\left(\mathbf{w}\mathbf{s}'\right) = (\mathbf{w}\mathbf{s}')\mathbf{w}_{\text{Linear3}}. \quad\quad (2)$$

**Complexity Analysis** We present a detailed complexity analysis of the original Physics-Attention mechanism in Table 1. Performing self-attention in the slice domain reduces the interaction complexity to $O(M^2C)$, which is quite efficient since $M \ll N$. However, the overall complexity is still fundamentally tethered to the mesh resolution $N$. For industrial-scale geometries with $N$ surpassing $10^8$, each $O(N)$ operation becomes a heavy bottleneck with high time complexity and massive memory footprint.

In a typical hyperparameter configuration of Transolver where $N \gg C \sim M$, the projections $\mathrm{Linear1}$ and $\mathrm{Linear3}$ pose the most significant scaling challenges for their $O(NC^2)$ time complexity and $O(NC)$ memory consumption. Meanwhile, the generation and storage of the slice weights $\mathbf{w}$ introduces a significant $O(NM)$ burden. These resolution-dependent operations seem to make it impossible for Transolver to process industrial-scale geometries, as caching an $N \times M$ tensor only necessitates prohibitive GPU memory, both during training and inference.

These bottlenecks necessitate a paradigm shift toward **geometry scaling** via structural optimizations and improved training and inference strategies in Transolver-3. In the next two subsections, we demonstrate how Transolver-3 systemati-

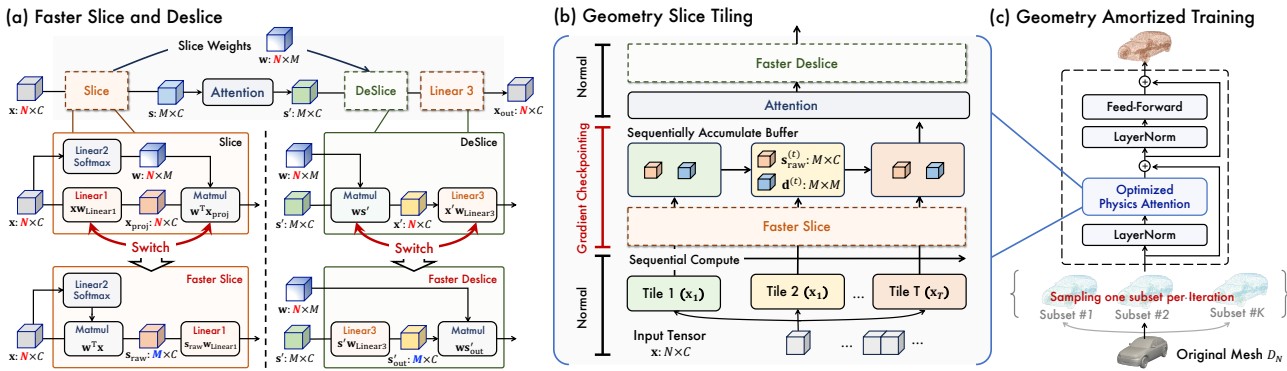

*Figure 2.* Geometry scaling at the training phase. (a) Comparison between the original Physics-Attention and its optimized version with faster slice and deslice. For brevity, we omit the diagonal matrix $\mathbf{d}$. (b) Geometry slice tiling partitions the computation of slice weights $\mathbf{w}$ to save memory. (c) Geometry amortized training allows learning on industrial-scale geometries with randomly sampled subsets.

*Table 1.* Complexity Analysis of Original Physics-Attention.

| Operation | Time Complexity | Space Complexity |
|---|---|---|
| Linear1($\mathbf{x}$) | $O(NC^2)$ | $O(NC)$ |
| Softmax(Linear2($\mathbf{x}$)) | $O(NCM)$ | $O(NM)$ |
| $(\mathbf{wd}^{-1})^\top \mathbf{x}_{\text{proj}}$ | $O(NMC)$ | $O(MC)$ |
| Attention($\mathbf{s}$) | $O(M^2C)$ | $O(M^2 + MC)$ |
| $\mathbf{ws}'$ | $O(NMC)$ | $O(NC)$ |
| Linear3($\mathbf{ws}'$) | $O(NC^2)$ | $O(NC)$ |
| $N$-**Related Terms** | 5 | 4 |

*Table 2.* Complexity Analysis of Optimized Physics-Attention.

| Operation | Time Complexity | Space Complexity |
|---|---|---|
| Softmax(Linear2($\mathbf{x}$)) | $O(NCM)$ | $O(NM)$ |
| $\mathbf{w}^\top \mathbf{x}$ | $O(NMC)$ | $O(MC)$ |
| Linear1($\mathbf{s}_{\text{raw}}$)$\mathbf{d}^{-1}$ | $O(MC^2)$ | $O(MC)$ |
| Attention($\mathbf{s}$) | $O(M^2C)$ | $O(M^2 + MC)$ |
| Linear3($\mathbf{s}'$) | $O(MC^2)$ | $O(MC)$ |
| $\mathbf{ws}'_{\text{out}}$ | $O(NMC)$ | $O(NC)$ |
| $N$-**Related Terms** | 3 | 2 |

cally resolve the scaling bottlenecks to enable high-fidelity simulations on industrial-scale geometries.

### 3.2. Geometry Scaling at the Training Phase

To bridge the gap between high-resolution data and limited GPU capacity, we propose a unified framework which optimizes both the model architecture and the training strategy, facilitating effective geometry scaling at the training phase.

**Faster slice and deslice**    As illustrated above, the fundamental scalability challenges of Physics-Attention resides in the operations with $O(N)$ complexity. In order to achieve optimal efficiency, we need to avoid all such operations if possible. As illustrated in Figure 2(a), we systematically reorder the operations in Physics-Attention to reduce $O(N)$ operations while maintaining computational equivalence. This is done by exploiting the matrix multiplication associative property. Specifically, the original slice and the deslice process are respectively optimized as:

$$\mathbf{s} = \text{Linear1}(\mathbf{w}^\top \mathbf{x})\mathbf{d}^{-1} = (\mathbf{w}^\top \mathbf{x})\mathbf{w}_{\text{Linear1}}\mathbf{d}^{-1},$$
$$\mathbf{x}_{\text{out}} = \mathbf{w}\text{Linear3}(\mathbf{s}') = \mathbf{ws}'\mathbf{w}_{\text{Linear3}}. \quad (3)$$

The faster slice and deslice operations are mathematically equivalent to the original processes in Eq (1) and (2). As detailed in the complexity analysis for the optimized Physics-Attention in Table 2, we successfully remove two of the five operations with $O(N)$ time complexity. Moreover, the opti-

mized architecture also helps alleviate memory bottleneck. In the original formalization, three intermediate tensors with $O(NC)$ footprint need to be cached for the backward pass. In the optimized version, two of these tensors are eliminated with the linear projections switched into the latent space.

**Geometry slice tiling**    While the faster slice and deslice operations have effectively eliminated two memory burdens, specifically the intermediate tensors $\mathbf{x}_{\text{proj}}$ and $\mathbf{x}'$ with $O(NC)$ memory requirements, the slice weights $\mathbf{w}$ still consume $O(NM)$ memory. Aside from the irreducible input tensors $\mathbf{x}$ and output tensors $\mathbf{x}_{\text{out}}$, this remains the largest intermediate bottleneck in the architecture. To address this, we introduce geometry slice tiling, a simple but effective memory-management strategy that eliminates the need to materialize the full $N \times M$ slice weight matrix in memory.

As described in Figure 2(b), the computation for physical states $\mathbf{s}$ is partitioned into $T$ tiles, each with tile size $N_t = \frac{N}{T}$. Each tile $\mathbf{x}_t$ produces its own contribution $\mathbf{s}_{\text{raw}}^{(t)}$ and $\mathbf{d}^{(t)}$ independently, which are then accumulated to the global $\mathbf{s}_{\text{raw}}$ and $\mathbf{d}$. This ensures that the full $N \times M$ slice weights matrix $\mathbf{w}$ is never materialized in memory. Afterwards, the global physical states are computed as

$$\mathbf{s} = \text{Linear1}\left(\sum_{t=1}^{T}((\mathbf{w}^{(t)})^\top \mathbf{x}^{(t)})\right)\left(\sum_{t=1}^{T}\mathbf{d}^{(t)}\right)^{-1}. \quad (4)$$

During training, the computation for each tile is wrapped

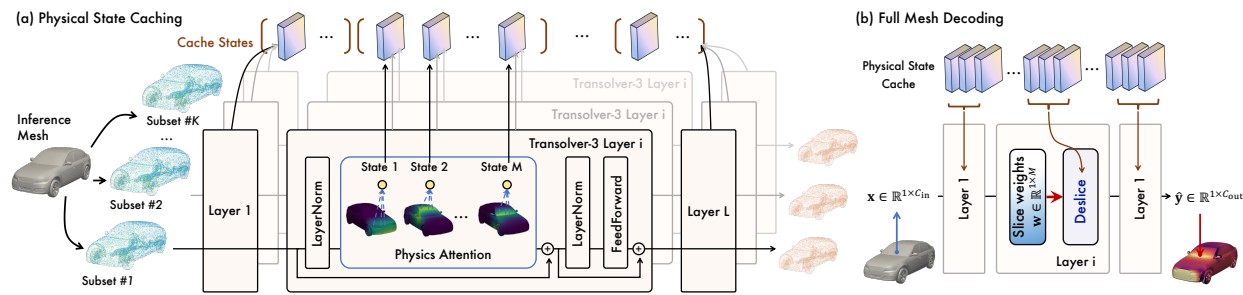

*Figure 3.* Decoupled inference of Transolver-3. (a) Physical state caching: global information is aggregated from the high-resolution mesh into cached states. (b) Full mesh decoding: physical fields on mesh coordinates are predicted by interacting with the physical state cache.

---

**Algorithm 1** Physics-Attention with Faster Slice and Deslice Mechanism and Geometry Slice Tiling

---

**Require:** Input features $\mathbf{x} \in \mathbb{R}^{N \times C}$, number of tiles $T$.
**Ensure:** Updated output features $\mathbf{x}_{\text{out}} \in \mathbb{R}^{N \times C}$.
1: Partition $\mathbf{x}$ into $T$ tiles $\{\mathbf{x}^{(t)}\}_{t=1}^{T}$ with size $N_t = N/T$.
2: Initialize global accumulators: $\mathbf{s}_{\text{raw}} \leftarrow \mathbf{0}, \mathbf{d} \leftarrow \mathbf{0}$.
3: **for** $k = 1$ **to** $T$ **do**
4:    $\mathbf{w}^{(t)} \leftarrow \text{Checkpoint}\left(\text{Softmax}(\text{Linear2}(\mathbf{x}^{(t)}))\right)$
5:    $\mathbf{s}_{\text{raw}}^{(t)} \leftarrow \text{Checkpoint}\left(\mathbf{w}^{(t)\top}\mathbf{x}^{(t)}\right)$
6:    $\mathbf{d}_{jj}^{(t)} \leftarrow \text{Checkpoint}\left(\sum_{i=1}^{N_t}\mathbf{w}_{ij}^{(t)}\right)$
7:    $\mathbf{s}_{\text{raw}} \leftarrow \mathbf{s}_{\text{raw}} + \mathbf{s}_{\text{raw}}^{(t)};\quad \mathbf{d} \leftarrow \mathbf{d} + \mathbf{d}^{(t)}$
8: **end for**
9: $\mathbf{s} \leftarrow \text{Linear1}(\mathbf{s}_{\text{raw}}\mathbf{d}^{-1})$
10: $\mathbf{s}'_{\text{out}} \leftarrow \text{Linear3}(\text{Attention}(\mathbf{s}))$
11: **for** $t = 1$ **to** $T$ **do**
12:    $\mathbf{x}_{\text{out}}^{(t)} \leftarrow \mathbf{w}^{(t)}\mathbf{s}'_{\text{out}}$
13: **end for**
14: Concatenate all tiles $\mathbf{x}_{\text{out}}^{(t)}$ to get $\mathbf{x}_{\text{out}}$
15: **return** $\mathbf{x}_{\text{out}}$

---

in gradient checkpointing to further reduce memory consumption. By *trading computation for memory*, geometry slice tiling compresses the peak memory consumption of $\mathbf{w}$ from $O(NM)$ to $O(N_t M)$, eliminating the last intermediate operations in Physics-Attention with $O(N)$ memory. The combined optimizations of the faster slice and deslice mechanism and geometry slice tiling enable near-optimal memory efficiency of Transolver-3, paving the way for geometry scaling to meet industrial requirements. Algorithm 1 provides the pseudo-code for these two optimizations.

**Geometry amortized training** Empowered with aforementioned optimization techniques, Transolver-3 is able to process around 2.9M cells on a single GPU. However, processing meshes with over $10^8$ is still computationally prohibitive. From the deep learning perspective, high-fidelity computational meshes serve as discrete samplings of the underlying continuous manifold. As Transolver is designed to learn intrinsic physical states from this continuous domain, the computational burden of processing the industrial-scale mesh can be amortized across different training steps by randomly sampling subsets from the complete mesh.
In each iteration, rather than processing the full mesh $D_N$,

the model is trained on a random subset $D_n$, where the subset with size $n \sim 10^5 - 10^6$ is uniformly sampled without replacement from the total $N$ mesh cells. Although a reduced sampling density may introduce discretization error, it does not impede the fundamental ability of the model to approximate the continuous operator. Meanwhile, by exposing the model to varying geometry subsets, the model is forced to learn the underlying physics laws on the continuous geometry manifold without relying on a certain form of discretization, further boosting its robustness.

### 3.3. Geometry Scaling at the Inference Phase

Even at the inference phase, processing the full mesh is still memory prohibitive. We identify that the physical state estimation and the field decoding processes can essentially be decoupled, and propose a two-stage inference framework comprising physical state caching and full mesh decoding, enabling high-fidelity prediction on industrial-scale geometries, achieving geometry scaling at the inference phase.

**Physical state caching** While processing the full mesh $D_N$ is prohibitive during training, the Transolver architecture allows for full utilization of information from the high-fidelity mesh during inference, regardless of the mesh size. As noted in Eq (4), the computation of physical states is inherently parallelizable and can be partitioned across independent chunks. Consequently, Transolver-3 is capable of ingesting meshes of arbitrary sizes by partitioning the input into memory-compatible chunks and aggregating the information together. The result is a global physical state cache which can be used for downstream physical field prediction.

As shown in Figure 3, the physical state cache for a high-fidelity mesh is constructed layer-by-layer. Suppose the input feature to layer $l$ is $\mathbf{x}^{(l)} \in \mathbb{R}^{N \times C}$, where $N$ represents the full size of the mesh, potentially exceeding the memory capacity of a single-pass forward. $\mathbf{x}$ is first partitioned into $K$ non-overlapping chunks $\{\mathbf{x}_k\}_{k=1}^{K}$ so that each chunk fits into the memory. The physical states $\mathbf{s}^{(l)}$ of layer $l$ is calculated by accumulating the contributions of each chunk, similar to Eq (4), ensuring estimating physical states with full geometry information. To facilitate prediction, we choose to cache $\mathbf{s}_{\text{out}}^{\prime(l)}$ in Figure 2(a) for better efficiency.

The resulting $\mathbf{s}'^{(l)}_{\text{out}}$ is exactly the same as calculating it in a single forward pass on a GPU with unlimited memory. The physical states of each layer constructs the physical state cache together, denoted as $\mathbf{s}_{\text{cache}} = \{\mathbf{s}'^{(l)}_{\text{out}}\}_{l=1}^{L}$, which can then be used to decode the full prediction results on $D_N$.

**Full mesh decoding**  With the physical state cache $\mathbf{s}_{\text{cache}}$, the predictions on each coordinate of the mesh can be derived. As shown in Figure 3, to get the prediction for coordinate $\mathbf{x} \in \mathbb{R}^{1 \times C_{\text{in}}}$, we need to update its feature layer by layer by interacting with $\mathbf{s}_{\text{cache}}$. The calculation is formalized as:

$$
\begin{aligned}
\mathbf{w}^{(l)} &= \text{Softmax}(\text{Linear2}(\mathbf{x}^{(l)})), \\
\mathbf{x}^{(l)}_{\text{out}} &= \mathbf{w}^{(l)} \mathbf{s}'^{(l)}_{\text{out}}.
\end{aligned}
\tag{5}
$$

Here $\mathbf{s}'^{(l)}_{\text{out}}$ is the $l$-th cached physical state. This formulation allows the model to decode the physical field at any spatial point on mesh $D_N$ with minimal incremental computation, as the most computationally intensive operations to calculate the physical state cache are performed only once.

*Remark 3.1* (**Importance of geometry scaling.**) Critical engineering quantities such as drag and lift coefficients are calculated as integrals over the surface $S$ of dimension $C_{\text{in}}$. A neural PDE solver approximates this continuous integral via numerical quadrature over a finite set of $N_s$ sampled points, and the total estimation error comprises the model's approximation error and the numerical quadrature error. Let $\hat{\mathcal{F}}$ be the learned physics field, the quadrature error is bounded by cell size $h \propto N_s^{-1/C_{\text{in}}}$ (Ferziger et al., 2019):

$$
\left\| \oint_S \hat{\mathcal{F}}(x)\mathrm{d}S - \sum_{i=1}^{N_s} \hat{\mathcal{F}}(x_i)\Delta S_i \right\| \leq h^{\alpha} \propto C N_s^{-\alpha/C_{\text{in}}}, \tag{6}
$$

where $C$ is a constant related to the field's derivatives and $\alpha > 0$ depends on the quadrature scheme. This highlights that high-fidelity prediction is critical for the accuracy of engineering quantities on industrial-scale geometries.

## 4. Experiments

We conduct comprehensive experiments on three industrial-level benchmarks featuring meshes with up to 160M cells, evaluating both point-wise prediction error and downstream design-oriented metrics such as drag and lift coefficients.

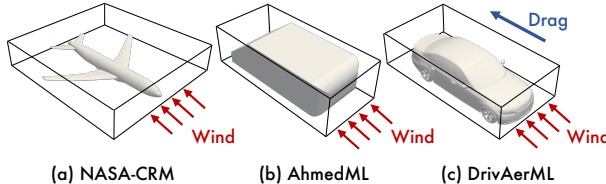

*Figure 4.* Car and aircraft simulation tasks, focusing on predictions of physical fields on the vehicle surface as well as surround volume.

**Benchmarks**  As summarized in Table 3, our benchmarks focus on high-fidelity aerodynamic simulation datasets that are representative of real-world industrial applications. Specifically, we conduct experiments on three public datasets: NASA-CRM (Bekemeyer et al., 2025), AhmedML (Ashton et al., 2024a) and DrivAerML (Ashton et al., 2024b). AhmedML and DrivAerML are automotive aerodynamics benchmarks featuring parametrically varied vehicle geometries with high-resolution Hybrid RANS-LES simulations, while NASA-CRM targets aircraft aerodynamics based on the widely studied Common Research Model. These datasets cover a wide range of mesh resolutions, spanning from $10^5$ to $1.6 \times 10^8$ (160M) mesh elements. Across all benchmarks, models are trained to perform full field prediction, including surface and volume variables.

*Table 3.* Summary of benchmarks, where #Mesh denotes the size of mesh cells per sample, and Size denotes the disk storage footprint.

| BENCHMARKS | GEO TYPE | #MESH | SIZE |
|---|---|---|---|
| NASA-CRM | SURFACE | ~400K | ~4GB |
| AHMEDML | SURFACE VOLUME | ~1M ~20M | ~8TB |
| DRIVAERML | SURFACE VOLUME | ~8M ~160M | ~31TB |

**Baselines**  We conduct comprehensive comparisons between Transolver-3 against 7 representative baseline models spanning different classes of neural PDE solvers. These baselines include graph-based deep architectures, such as Graph U-Net (Gao & Ji, 2019) and MeshGraph-Net (Sanchez-Gonzalez et al., 2020), neural operator families, such as GINO (Li et al., 2023b), and transformer-based solvers designed for irregular meshes, such as GAOT (Wen et al., 2025). Notably, UPT (Alkin et al., 2024) and its adaptive variant AB-UPT (Alkin et al., 2025) provide scalable frameworks for high-fidelity CFD on unstructured meshes. We also include Transolver and its successor Transolver++.

### 4.1. Main Results

We demonstrate the efficacy of Transolver-3 against state-of-the-art baselines across three well-established aerodynamic benchmarks: complex aircraft geometries, bluff-body flows (AhmedML) and high-fidelity automotive simulations (DrivAerML), providing a comprehensive assessment of the model's performance across varied geometric complexities.

As shown in Table 4, Transolver-3 outperforms all baselines on 9 out of 10 metrics. The sole exception is the volume pressure $\boldsymbol{p}_v$ on AhmedML, where Transolver-3 lags behind AB-UPT by only a small margin. The performance gain of Transolver-3 over Transolver++ is primarily due to the architectural refinement of the projection layers. While Transolver++ removes the Linear1 layer from the original

*Table 4.* Relative L2 errors (in %) of surface pressure $p_s$ and skin friction coefficient $C_f$ on the NASA-CRM dataset, and surface pressure $p_s$, volume velocity $u$, wall shear stress $\tau$ and volume pressure $p_v$ on the AhmedML and DrivAerML datasets.

| MODELS | NASA-CRM | | AHMEDML | | | | DRIVAERML | | | |
|---|---|---|---|---|---|---|---|---|---|---|
| | $p_s$ | $C_f$ | $p_s$ | $u$ | $\tau$ | $p_v$ | $p_s$ | $u$ | $\tau$ | $p_v$ |
| GRAPH U-NET* | 15.85 | 15.61 | 6.46 | 4.15 | 7.29 | 5.18 | 16.13 | 17.98 | 27.84 | 20.51 |
| GINO* | 12.39 | 11.51 | 7.90 | 6.23 | 8.18 | 8.80 | 13.03 | 40.58 | 21.71 | 44.90 |
| MESHGRAPHNET* | 14.80 | 9.81 | 3.72 | 3.97 | 5.01 | 4.50 | 14.30 | 15.01 | 20.13 | 21.82 |
| GAOT* | 30.38 | 59.79 | 8.02 | 7.43 | 9.92 | 10.47 | 34.00 | 57.18 | 61.00 | 56.90 |
| UPT | 12.78 | 23.78 | 4.25 | 2.73 | 5.80 | 3.10 | 7.44 | 8.74 | 12.93 | 10.05 |
| AB-UPT | 9.77 | 6.43 | 3.97 | 1.94 | 5.60 | **2.07** | 3.82 | 5.93 | 7.29 | 6.08 |
| TRANSOLVER* | 9.61 | 7.04 | 3.20 | 1.81 | 4.85 | 2.41 | 4.81 | 6.78 | 8.95 | 7.74 |
| TRANSOLVER++* | 9.51 | 6.95 | 3.47 | 1.78 | 5.06 | 2.35 | 4.12 | 4.70 | 6.42 | 6.70 |
| **TRANSOLVER-3** | **8.71** | **5.85** | **2.96** | **1.60** | **4.81** | 2.16 | **3.71** | **4.14** | **5.85** | **5.72** |

\* The input mesh sizes on AhmedML and DrivAerML may exceed the maximum input capacity of these models. To enable comparison, we apply the geometry amortized training strategy. At the evaluation phase, we split the input mesh into several chunks and perform inference on these chunks independently. The outputs from these chunks are concatenated to form the full results.

Physics-Attention mechanism to reduce memory consumption, Transolver-3 preserves this linear layer and shifts it into the slice domain. This preserves the model's full expressive capacity while maintaining high memory efficiency. Notably, in the DrivAerML benchmark, Transolver-3 successfully handles over **160 million cells** of the volume field, surpassing competing models by a significant margin.

We observe that while GNN-based models like Graph U-Net and GINO performs reasonably well on modest-scale benchmarks like NASA-CRM, their performance degrades significantly on larger benchmarks like DrivAerML. This performance drop highlights their lack of scalability when transitioning to large-scale simulation scenarios, where the increased geometric complexity and mesh resolution pose substantial challenges for graph-based architectures. We also evaluate GAOT, a recently published neural PDE solver which utilizes a graph-based encoder-decoder with an intermediate Vision Transformer. We observe prohibitive memory consumption when trying to reproduce its results on high-resolution meshes, and it also exhibits sub-optimal performance on both NASA-CRM and DrivAerML compared to other baselines. As a representative scalable solver, AB-UPT builds upon UPT and introduces several architectural refinements, showing robust performance across all three benchmarks. However, Transolver-3 still consistently outperforms AB-UPT on 9 out of 10 metrics, demonstrating its state-of-the-art performance on industrial-scale simulations.

**Prediction of integrated quantities** Integrated quantities, such as the drag coefficient ($C_d$) or lift coefficient ($C_l$), are pivotal for practical industrial applications. We evaluate the performance of Transolver-3 in predicting these quantities on DrivAerML, the largest-scale dataset in our study featuring industrial-level mesh fidelity. Since these coefficients are derived by integrating surface pressure $p_s$ and wall shear stress $\tau$ across all mesh cells of the vehicle surface, the capacity for full-field prediction on industrial-scale meshes

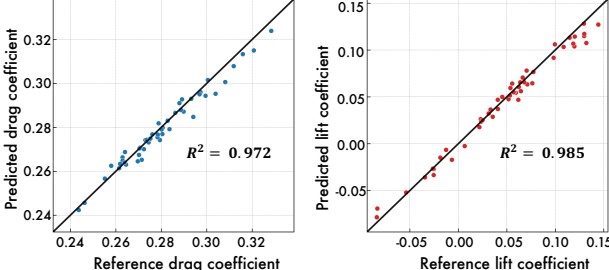

*Figure 5.* Transolver-3 can produce accurate predictions of drag and lift coefficients on the DrivAerML dataset.

is essential for producing accurate results. As shown in Figure 5, Transolver-3 achieves exceptional coefficients of determination ($R^2$) on $C_d$ and $C_l$ predictions, surpassing the results achieved by AB-UPT (0.963 for $C_d$ and 0.975 for $C_l$). These precise predictions underscore the capability of Transolver-3 in handling industrial-scale simulations, making it highly suitable for real-world design workflows.

### 4.2. Model Analysis

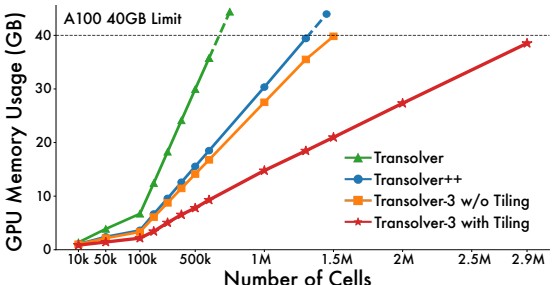

*Figure 6.* Comparison of single-GPU capacity without amortized training. Transolver-3 achieves 1.9×capacity of Transolver++, serving as the foundation for scaling to over $10^8$ cells.

**Memory footprint analysis** While geometry amortized training allows training on subsets of the full mesh, maximizing the single-GPU capacity remains critical for preserving high-resolution physical details during training. We

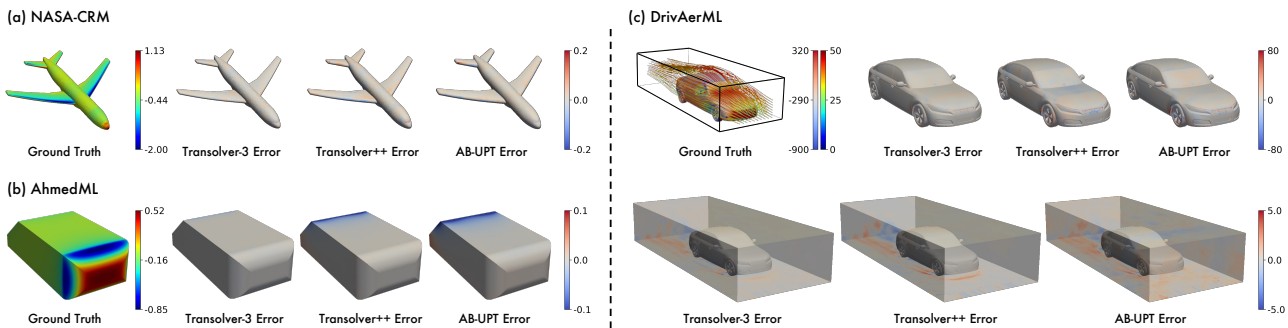

*Figure 7.* Case study of error maps on different benchmarks. We plot the error maps of surface pressure on NASA-CRM and AhmedML. For DrivAerML, we plot the surface pressure and the volume velocity simultaneously. See Appendix C.2 for more showcases.

evaluate the impact of faster slice and deslice mechanism and geometry slice tiling on memory reduction, with the results presented in Figure 6. All memory consumptions are evaluated during the training phase using identical hyperparameter configurations across models. Transolver-3 with faster slice and deslice can handle around 10% more mesh cells than Transolver++. Notably, Transolver++ omits $\mathrm{Linear}1$ layer to save memory, potentially compromising model capacity, while Transolver-3 retains full capacity of Physics-Attention. Furthermore, as shown in Figure 6, integrating geometry slice tiling boosts single-GPU capacity by roughly 90%. Note that the 2.9M limit here represents the raw capacity for single-pass processing. Further integration with amortized training and decoupled inference facilitates scaling to industrial geometries exceeding $10^8$ cells.

*Table 5.* Memory consumption and latency with different tile sizes, both evaluated at the training phase. The total input size is set to 800k, which is then partitioned with different tile sizes.

| TILE SIZE | 800k | 200k | **100k** | 20k | 10k | 5k |
|---|---|---|---|---|---|---|
| NUM OF TILES | 1 | 4 | **8** | 40 | 80 | 160 |
| MEMORY (GB) | 22.16 | 13.62 | **12.28** | 11.27 | 11.09 | 11.04 |
| LATENCY (MS) | 404 | 410 | **429** | 453 | 656 | 1307 |

However, this optimization adheres to the "no free lunch" principle, as it essentially trades computation for memory. As illustrated in Table 5, while memory usage decreases with small tile size, the training latency increases as well. We observe that a tile size of 100k serves as an ideal choice in this setting, which does not induce much computational burden but significantly reduces the memory consumption. Therefore, the selection of tile size must strike a balance between practical time constraints and memory availability.

**Time complexity analysis** Beyond memory efficiency, Transolver-3 significantly enhances computational speed as well. As illustrated in Figure 8, we evaluate both the theoretical GFLOPs and the practical hardware latency of the Physics-Attention mechanism. While Transolver-3 achieves a 20% reduction in GFLOPs, the practical impact is more significant: the latency decreases by approximately 60% due to the optimization of operation orders via faster slice

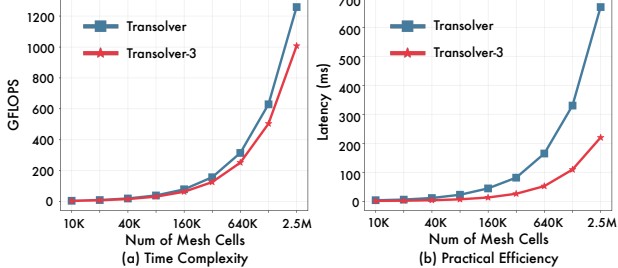

*Figure 8.* Time complexity analysis of Transolver-3. (a) GFLOPs of Transolver and Transolver-3 under varying mesh cells. (b) Practical inference latency measured on A100 GPU, showing that Transolver-3 achieves substantially lower latency.

and deslice. By migrating the projection layers from the high-dimensional mesh domain to compact slice domain, we reduce redundant memory access and avoid the generation of massive intermediate tensors. The reduced computation time facilitates faster predictions on industrial-scale meshes, further boosting Transolver-3's practical applicability.

**Impact of geometry scaling** We evaluate the impact of geometry scaling during both physical state caching and full mesh decoding, where we scale up the input and evaluation resolution, respectively. As shown in Figure 9(a), the prediction error decreases consistently as the number of cells utilized for the physical state cache approaches the full mesh resolution. Meanwhile, Figure 9(b) reveals that the evaluation resolution is critical for the precision of integrated quantities; specifically, increasing the density of sampled surface cells consistently refines the accuracy of drag and lift coefficients. These results underscore that scaling of input and evaluation resolutions is indispensable for achieving the high-fidelity results required in industrial-scale simulations.

**Case study** As shown in Figure 7, Transolver-3 maintains a smaller and more uniform error distribution than Transolver++ and AB-UPT across all three benchmarks. In the NASA-CRM case, Transolver-3 demonstrates better accuracy in regions with high geometric curvature, such as the leading edges of the wings and tail. For the AhmedML benchmark, Transolver-3 also shows significantly better

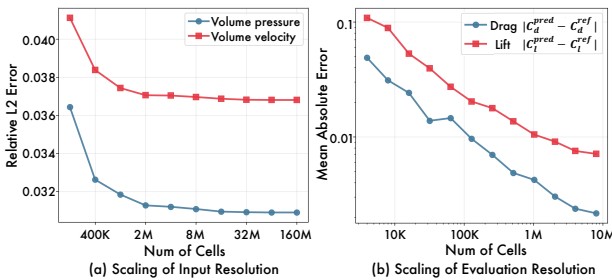

*Figure 9.* Impact of Geometry scaling. (a) With more input cells fed into the physical state cache, the prediction error consistently decreases. (b) The estimation of critical engineering quantities like $C_d$ benefits from higher evaluation resolutions.

performance in the rear regions. In the highly complex DrivAerML case, Transolver-3 delivers exceptionally precise surface pressure and volumetric velocity predictions. Notably, it yields superior results in the most challenging zones, including the vehicle front and the underbody area.

**Ablation on training subset size**    To investigate how the training subset size $n$ influences the model performance, we evaluate varying scales of $n$ on the DrivAerML dataset. The results are summarized in Table 6. The results reveal that while the prediction accuracy scales positively with larger subset sizes, the model exhibits robustness even when $n$ is highly constrained. Notably, even at the minimum tested subset size ($n = 1000$), the predictive performance degrades gracefully without encountering training instability or catastrophic collapse. This demonstrates that the amortized training strategy allows Transolver-3 to capture the underlying physics invariant to specific discretization densities, enabling practical trade-offs between computational overhead and target accuracy on industrial-scale geometries.

*Table 6.* Impact of training subset size $n$ on the DrivAerML dataset. Relative L2 errors (in %) are reported.

| SUBSET SIZE | $p_s$ | $u$ | $\tau$ | $p_v$ |
|---|---|---|---|---|
| 1k | 4.84 | 5.30 | 7.31 | 7.94 |
| 5k | 4.05 | 4.63 | 6.39 | 6.53 |
| 10k | 3.86 | 4.49 | 6.22 | 6.19 |
| 20k | 3.80 | 4.29 | 6.08 | 5.90 |
| 50k | 3.76 | 4.21 | 6.02 | 5.79 |
| 100k | 3.71 | 4.14 | 5.85 | 5.72 |

**PDEs with discontinuities or shocks**    To evaluate the performance of Transolver-3 on PDEs involving discontinuities, shocks, or more complex patterns, we conduct experiments on the AirCraft dataset. This dataset is proposed by Transolver++ (Luo et al., 2025) and involves transonic and supersonic flows with clear shocks and discontinuities. The performance comparison of Transovler-3 with AB-UPT and other Transolver models is shown in Table 7.

The AirCraft dataset contains around 300k points per sample, and we employ a subset size of 100k when training

*Table 7.* Performance comparison on the Aircraft dataset. Relative L2 errors (in %) of surface pressure $p_s$ is reported.

| VARIABLE | AB-UPT | Transolver | Transolver++ | Transolver-3 |
|---|---|---|---|---|
| $p_s$ | 6.93 | 9.20 | 6.40 | **5.80** |

Transolver-3. Transolver-3 outperforms both Transolver++ and AB-UPT on this challenging dataset, demonstrating that there are no architectural barriers to Transolver-3 handling systems involving discontinuities or shocks.

**Practical efficiency comparison**    We provide a detailed comparison of memory usage, training cost, and inference latency against AB-UPT on DrivAerML, as shown in Table 8. Transolver-3 achieves competitive computational efficiency. Despite a slightly higher training memory footprint, its training speed is approximately 10% faster. Furthermore, while there is a marginal increase in inference latency, Transolver-3 significantly reduces inference memory consumption, which is due to the decoupled inference strategy. This memory saving underpins its high practical utility for resource-constrained deployments, while the slight latency increase is justified by its superior predictive accuracy.

*Table 8.* Training and inference cost comparison between Transolver-3 and AB-UPT on DrivAerML. Memory usage (in GB), total training time (in GPU hours), and inference latency (in seconds) on a single A100 GPU are reported.

| MODELS | TRAINING | | INFERENCE | |
|---|---|---|---|---|
| | Memory | GPU hours | Memory | Latency (s) |
| AB-UPT | 10.0 | 40 | 11.2 | 220 |
| TRANSOLVER-3 | 12.5 | 36 | 6.0 | 260 |

## 5. Conclusions

This paper presents Transolver-3, a comprehensive framework designed to realize practical neural PDE solvers for industrial-scale geometries exceeding $10^8$ cells. We systematically resolve the memory bottleneck of geometry scaling across training and inference. During training, we bridge the gap between high-fidelity geometry representations and limited GPU memory through the integration of two structural optimizations and geometry amortized training. During inference, we introduce a decoupled framework that separates physical state caching from full mesh decoding, enabling the aggregation of global mesh information to achieve superior predictive performance. By successfully scaling to 160 million mesh cells, Transolver-3 demonstrates the potential to be integrated into real-world AI-CAE workflows. A promising direction is to further enhance the model's geometric understanding by incorporating structural priors from geometry pre-training frameworks, such as GeoPT (Wu et al., 2026). Integrating such geometric knowledge could potentially bolster Transolver-3's generalization capabilities across cross-domain industrial topologies, paving the way toward a geometry-aware physics foundation model.

## Acknowledgement

This work was supported in part by the Beijing Scholar Program, the Fundamental and Interdisciplinary Disciplines Breakthrough Plan of the Ministry of Education of China (JYB2025XDXM803), the Natural Science Foundation of China (U2342217), and the National Engineering Research Center for Big Data Software.

## Impact Statement

This paper presents the work whose goal is to advance the field of neural PDE solvers for high-fidelity industrial applications. The proposed method is based on a detailed analysis of the Physics-Attention mechanism, and proposes several architectural optimizations toward better memory efficiency and faster computation, as well as practical training and inference strategies to enable prediction on industrial-scale geometries. Note that this paper primarily focuses on the scientific and engineering problem. Thus we believe that there are no potential ethical risks related to our work.

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

# A. Implementation Details

## A.1. Benchmarks

**NASA Common Research Model.** The NASA Common Research Model (CRM) benchmark (Bekemeyer et al., 2025) provides high-fidelity CFD simulations for a full wing–body–horizontal tail transport aircraft configuration under realistic 1g flight shape deformation. Simulations were performed using the DLR TAU RANS solver with the Spalart–Allmaras turbulence model. Six input parameters are considered, including Mach number, angle of attack, and control surface deflections. The altitude is fixed at 37,000 ft. The outputs include distributed surface pressure and skin friction coefficients, surface geometric information, as well as integrated aerodynamic coefficients. The dataset contains 149 valid CFD samples and provides a predefined split of 105 training samples and 44 testing samples. Our experimental setup follows this protocol.

**AhmedML.** AhmedML (Ashton et al., 2024a) is a publicly available dataset offering high-fidelity CFD results for 500 different geometric configurations of the Ahmed bluff body, a classic benchmark in automotive aerodynamics research. The simulations employ hybrid RANS-LES turbulence modeling in OpenFOAM and accurately resolve key flow features, including pressure-driven separation and three-dimensional vortical structures. Each case uses a mesh of approximately 20 million cells. As no official train/validation/test split is provided, we randomly partition the dataset into 400 training samples, 50 validation samples, and 50 test samples. Our models are trained to predict surface pressure and wall shear stress on the mesh, as well as velocity and pressure fields.

**DrivAerML.** DrivAerML (Ashton et al., 2024b) is a dataset developed specifically to support machine learning applications in high-fidelity automotive aerodynamics. It consists of 500 parametrically deformed variants of the DrivAer vehicle model, created to help overcome the scarcity of large-scale, open-source CFD data in this field. The simulations are conducted using hybrid RANS-LES methods (Spalart et al., 2006; Chaouat, 2017; Heinz, 2020; Ashton et al., 2022) on volumetric meshes of approximately 140 million cells—the highest-fidelity approach commonly employed in the automotive industry (Ma et al., 2025; Ashton et al., 2024b). Each surface mesh contains roughly 8.8 million points, providing pressure and wall shear stress on the surface, along with velocity, pressure, and vorticity throughout the volume. Drag and lift coefficients are computed using all 8.8 million surface points in the Transolver-3 method. Since no predefined data split is available, we randomly allocate 400 samples for training and 50 for testing, while reserving 50 for validation. Of these validation cases, 16 lack simulation results and are treated as hidden samples, yielding 34 effective validation samples.

## A.2. Metrics

To comprehensively evaluate model performance across different datasets and prediction tasks, we adopt the relative L2 error as the primary evaluation metric for point-wise flow field predictions. This metric is applied to key physical quantities, including surface pressure coefficients($p_s$), wall shear stress ($\tau$), and volume fields (velocity($u$) and pressure($p_v$)). For the prediction of integrated aerodynamic coefficients, which are critical downstream engineering quantities, we report the R-squared (R²) score to assess the accuracy and explanatory power of the predicted drag ($C_D$) and lift ($C_L$) coefficients. In experiments specifically focused on geometry scaling and large-scale industrial meshes, where precise quantification of global performance under memory constraints is emphasized, we additionally employ the mean absolute error (MAE) for the integrated coefficients. These metrics collectively provide a robust assessment of both local field fidelity and global aerodynamic accuracy across the NASA-CRM, AhmedML, and DrivAerML benchmarks.

**Relative L2 Error**

In this work, we primarily report the relative $L_2$ error as the main metric for evaluating point-wise flow field predictions. Given a high-resolution mesh with $N$ points (or cells), the relative $L_2$ error for a single sample compares the predicted field $\hat{\mathbf{Y}} \in \mathbb{R}^{N \times d_{\text{out}}}$ against the ground-truth high-fidelity CFD solution $\mathbf{Y} \in \mathbb{R}^{N \times d_{\text{out}}}$, where $d_{\text{out}}$ denotes the dimensionality of the target output. It is defined as:

$$\text{L2}_{\text{rel}}(\mathbf{Y}, \hat{\mathbf{Y}}) = \frac{\|\hat{\mathbf{Y}} - \mathbf{Y}\|_2}{\|\mathbf{Y}\|_2} = \frac{\sqrt{\sum_{n=1}^{N} \sum_{d=1}^{d_{\text{out}}} (\hat{\mathbf{Y}}_{n,d} - \mathbf{Y}_{n,d})^2}}{\sqrt{\sum_{n=1}^{N} \sum_{d=1}^{d_{\text{out}}} \mathbf{Y}_{n,d}^2}}. \tag{7}$$

For a test set containing $M$ samples, with ground-truth fields $\mathbf{Y} = \{\mathbf{Y}^{(1)}, \mathbf{Y}^{(2)}, \ldots, \mathbf{Y}^{(M)}\}$ and corresponding predictions $\hat{\mathbf{Y}} = \{\hat{\mathbf{Y}}^{(1)}, \hat{\mathbf{Y}}^{(2)}, \ldots, \hat{\mathbf{Y}}^{(M)}\}$, the mean relative $L_2$ error is computed as:

$$\text{L2}_{\text{rel}}(\mathbf{y}, \hat{\mathbf{y}}) = \frac{1}{M} \sum_{m=1}^{M} \text{L2}_{\text{rel}}(\mathbf{Y}^{(m)}, \hat{\mathbf{Y}}^{(m)}). \tag{8}$$

When reporting field-specific results, the relative $L_2$ error is calculated separately for each physical quantity. During training, input features are preprocessed according to their type. Geometric features, such as point coordinates, are typically first normalized using min-max scaling and then optionally multiplied by a constant scaling factor (e.g., 1000) to preserve numerical stability and relative spatial relationships. The target outputs, on the other hand, are standardized to have zero mean and unit variance across the dataset. This preprocessing strategy stabilizes optimization and ensures consistent training.

This metric on all benchmarks is reported in the Table 4, providing a direct comparison across different models.

**Integrated Aerodynamic Coefficients ($C_d$ and $C_l$)**

In addition to point-wise flow field predictions, accurate estimation of integrated aerodynamic coefficients is crucial for downstream engineering applications, such as aircraft loads analysis and vehicle performance optimization. These global quantities—drag coefficient ($C_d$) and lift coefficient ($C_l$)—are derived from surface integration of the predicted pressure and wall shear stress fields over the full high-resolution surface mesh.

In aerodynamics, the total force $\mathbf{F}$ acting on an object is defined as the surface integral:

$$\mathbf{F} = \int_S \left[ -(p(\mathbf{x}) - p_\infty)\mathbf{n}(\mathbf{x}) + \boldsymbol{\tau}(\mathbf{x}) \right] \, dS, \tag{9}$$

where $p$ is the surface pressure, $p_\infty$ is the freestream reference pressure, $\mathbf{n}$ is the outward unit normal vector, and $\boldsymbol{\tau}$ is the wall shear stress tensor.

The drag coefficient $C_d$ and lift coefficient $C_l$ are then normalized forms of the drag and lift components of $\mathbf{F}$:

$$C_d = \frac{\mathbf{F} \cdot \hat{\mathbf{d}}}{\frac{1}{2}\rho_\infty v_\infty^2 A}, \quad C_l = \frac{\mathbf{F} \cdot \hat{\mathbf{l}}}{\frac{1}{2}\rho_\infty v_\infty^2 A}, \tag{10}$$

where $\hat{\mathbf{d}}$ and $\hat{\mathbf{l}}$ are unit vectors in the drag and lift directions, $\rho_\infty$ and $v_\infty$ are freestream density and velocity, and $A$ is a reference area.

In our benchmarks, both $C_d$ and $C_l$ are evaluated for all cases. For automotive configurations (AhmedML and DrivAerML), these metrics quantify aerodynamic resistance and lift effects relevant for fuel efficiency, stability, and performance, while for the aircraft configuration (NASA-CRM), they assess the overall aerodynamic behavior across the flight envelope.

**R-squared ($R^2$) Score for Coefficient Predictions**

To further evaluate the model's ability to predict integrated aerodynamic coefficients across the test set, we report the R-squared ($R^2$) score as a complementary metric. For a set of $M$ samples with ground-truth coefficients $y_i$ and predictions $\hat{y}_i$ ($i = 1, \ldots, M$), the $R^2$ score is defined as:

$$R^2 = 1 - \frac{\sum_{i=1}^{M}(y_i - \hat{y}_i)^2}{\sum_{i=1}^{M}(y_i - \bar{y})^2}, \tag{11}$$

where $\bar{y}$ is the mean of the ground-truth values.

An $R^2$ score close to 1 indicates that the model explains nearly all variance in the target coefficients, reflecting strong generalization and physical consistency. Lower or negative values suggest poor predictive quality relative to a constant mean predictor. This metric is particularly valuable for assessing global performance on industrial-scale benchmarks, where precise coefficient prediction directly impacts design decisions.

**Mean Absolute Error (MAE) for Integrated Coefficients**

In experiments focused on geometry scaling—particularly when evaluating the impact of varying input and output resolutions on industrial-scale meshes—we employ the mean absolute error (MAE) to quantify the accuracy of predicted integrated

aerodynamic coefficients ($C_d$ and $C_l$). This metric is especially suitable for assessing global performance under resolution constraints.

For a test set of $M$ samples with ground-truth coefficients $y_i$ and model predictions $\hat{y}_i$ ($i = 1, \ldots, M$), MAE is defined as:

$$\text{MAE} = \frac{1}{M} \sum_{i=1}^{M} |y_i - \hat{y}_i|. \tag{12}$$

Lower MAE values indicate higher predictive precision for these design-critical quantities. As highlighted in our analysis (Figure 9 (b)), the evaluation resolution of the predicted solution field is critical for the precision of integrated quantities such as drag and lift coefficients. Increasing the number of sampled surface cells during inference consistently reduces MAE on $C_d$ and $C_l$, demonstrating that scaling both training/input resolution and evaluation/output resolution is essential for achieving high-fidelity results in industrial-scale simulations.

This metric complements relative $L_2$ errors on fields and $R^2$ scores on coefficients by providing an intuitive, scale-sensitive measure of global aerodynamic accuracy in resolution-ablation studies.

### A.3. Baselines and Implementations

As shown in Table 9, all the baselines are trained and tested under the same training strategy. Note that for relatively small-scale datasets like NARA-CRM, the geometry amortized training strategy is not applied, and all models are trained with the full mesh. For larger-scale benchmarks, geometry amortized training is applied to enable training of each model.

*Table 9.* Training and model configurations of Transolver-3. "Subset Size" denotes the size of the randomly sampled subsets in geometry amortized training strategy. "Full Mesh" means that we do not subsample on the benchmark for its relatively small mesh size.

| Benchmarks | Training Configuration (Shared in all baselines) | | | | | Model Configuration | | | |
|---|---|---|---|---|---|---|---|---|---|
| | Loss | Epochs | Initial LR | Optimizer | Subset Size | Layers $L$ | Heads | Channels $C$ | Slices $M$ |
| NASA-CRM | | | | | Full Mesh | 24 | | 256 | 64 |
| AhmedML | Relative L2 | 500 | $10^{-3}$ | AdamW | 100k | 16 | 8 | 256 | 64 |
| DrivAerML | | | | (2017) | 100k | 24 | | 256 | 64 |

It should be noted that the results reported in Table 4 for the AhmedML and DrivAerML benchmarks are primarily taken from the AB-UPT (Alkin et al., 2025) study, with the exception of Transolver and Transolver++, for which we provide our own experiments. Furthermore, as GAOT (Wen et al., 2025) did not report the Relative L2 Error on these benchmarks, we reproduced the results for GAOT to facilitate a direct comparison. Specifically, for each baseline model, we employed the AdamW optimizer (Loshchilov & Hutter, 2017), except for AB-UPT/UPT where LION (Chen et al., 2023) was used. For AdamW-trained models, the learning rate was selected via a sweep over $\{1 \times 10^{-3}, 5 \times 10^{-4}, 2 \times 10^{-4}, 1 \times 10^{-4}, 5 \times 10^{-5}, 2 \times 10^{-5}\}$ to achieve optimal performance for each baseline. Whenever LION was applied, a learning rate of $5 \times 10^{-5}$ was used, which has shown consistently good performance.

Training was conducted in either float16 or bfloat16 precision, depending on which yielded better results. GINO (Li et al., 2023b) was an exception, as it exhibited instability under mixed precision. All models were trained for 500 epochs with a batch size of 1, a weight decay of 0.05, and gradient clipping at 1. A cosine learning rate schedule was employed, including a 5% warm-up phase and a minimum learning rate of $1 \times 10^{-6}$.

For the NASA-CRM benchmark, the results were primarily obtained through our own re-implementation. In the following, we present the detailed implementation of all baselines.

**Graph-based Deep Architectures.**

We consider graph-based neural architectures that are specifically designed to operate on irregular discretizations, which naturally arise in unstructured meshes and point cloud representations commonly used in computational fluid dynamics.

**Graph U-Net.** Graph U-Net (Gao & Ji, 2019) is an extension of the classical U-Net architecture to graph-structured data. To enable hierarchical feature learning on irregular grids, Graph U-Net introduces two key operations: a graph pooling layer for downsampling and a graph unpooling layer for upsampling. These operations allow the model to preserve the encoder–decoder structure and skip connections of U-Net while operating directly on graph domains. In our experiments,

we adopt the public implementation provided by the Neural-Solver-Library (Wu et al., 2024b). The underlying graph connectivity is constructed using a $k$-nearest-neighbor (k-NN) graph with $k = 20$.

**GINO.** The Geometry-Informed Neural Operator (GINO) (Li et al., 2023b) is a neural operator framework that learns solution operators of large-scale partial differential equations by decoupling geometry encoding from field prediction. Similar to AB-UPT, GINO separates the processing of geometric information and physical fields. Specifically, GINO employs a Graph Neural Operator (GNO) to map irregular point cloud inputs to a regularly structured latent grid, enabling the efficient application of the Fourier Neural Operator (FNO) (Li et al., 2020a) in the latent space. After latent-space processing, a second GNO block acts as a field decoder to produce predictions at arbitrary query points on the original irregular domain, such as surface meshes of automotive geometries.To ensure a fair comparison with other baselines that rely solely on point cloud geometry, we remove the signed distance function (SDF) input feature from GINO in all experiments.

**Transformer-based Models.**

We further consider Transformer-based neural operators that leverage global self-attention mechanisms to capture long-range dependencies on irregular geometries and unstructured meshes.

**GAOT.** The Geometry-Aware Operator Transformer (GAOT) (Wen et al., 2025) is a neural PDE solver that combines a graph-based encoder–decoder with a Transformer-based global processor. Multi-scale Geometry-Aware Neural Operators (MAGNO) are used to encode irregular meshes into latent tokens and reconstruct continuous field predictions, while self-attention captures long-range interactions. Spatial coordinates are rescaled to $[-1, 1]^d$ and geometric information is embedded via local statistics. The Transformer uses a hidden dimension of 256, a feed-forward dimension of 1024, and 10 layers. For large-scale 3D datasets, latent token configurations of $[64, 64, 32]$ are used for NASA-CRM and AHMEDML, and $[64, 32, 32]$ for DrivAerML. Due to memory constraints on extremely high-resolution meshes, GAOT results are reported only for computationally feasible cases.

**UPT.** The Universal Physics Transformer (UPT) (Alkin et al., 2024) is a unified neural operator framework that does not rely on a grid- or particle-based latent structure. UPT operates directly on point cloud representations of the input function and supports querying the learned latent space at arbitrary spatial or spatio-temporal locations. Similar in spirit to GINO (Li et al., 2023b), UPT decouples input encoding from output querying through attention-based mechanisms. In UPT, the input point cloud is first mapped to a lower-dimensional representation via a message-passing supernode pooling layer, which reduces the number of tokens while preserving global contextual information. A stack of Transformer encoder blocks then processes the supernode representations to produce a compact latent embedding. To query the latent space at arbitrary locations, UPT employs cross-attention blocks between query points and the latent representation. While UPT optionally supports Perceiver-style pooling, we do not use it in our experiments, as the increased computational efficiency comes at the cost of reduced prediction accuracy, and our study prioritizes accuracy over efficiency. To ensure comparable computational budgets across Transformer-based baselines, we employ 12 cross-attention blocks instead of the single block used in the original configuration.

**AB-UPT.** The Anchored-Branched Universal Physics Transformer (AB-UPT) (Alkin et al., 2025) is a scalable Transformer-based neural surrogate architecture designed for high-fidelity CFD simulations on extremely large unstructured meshes, targeting industrial-scale automotive and aerospace aerodynamics with meshes exceeding $10^8$ cells. AB-UPT introduces a multi-branch architecture that explicitly decouples geometry encoding from field prediction. A dedicated geometry branch aggregates the input point cloud into a fixed set of geometry supernodes, providing a global structural representation of the underlying shape. In parallel, separate surface and volume branches are employed to predict field quantities on their respective domains, including pressure and wall shear stress on surfaces, and velocity, pressure, and vorticity in volumes.

The interaction between geometry, surface, and volume branches is enabled through a sequence of shared physics blocks, which employ cross-attention mechanisms both from field tokens to geometry tokens and between surface and volume branches. This design allows AB-UPT to jointly model coupled surface–volume dynamics while maintaining a clear separation between geometric and physical representations. Scalability is achieved via an anchored attention mechanism: a small, fixed set of anchor tokens performs self-attention to capture global physical interactions, while arbitrary query points—potentially corresponding to full-resolution surface or volume meshes—attend to these anchors via cross-attention. This formulation enables mesh-independent training and inference with constant memory consumption, making AB-UPT applicable to massive industrial geometries.

Since the official release of AB-UPT provides inference code only, we reimplemented the training pipeline to ensure consistency with our experimental setup. For the NASA-CRM and AhmedML benchmarks, models are trained for 2000

epochs with a batch size of 1, using the AdamW optimizer with a learning rate of $5 \times 10^{-4}$, weight decay 0.01, and cosine decay scheduling. We use a hidden dimension of 64, geometry depth of 2, 4 attention heads, and a block sequence `psssss`, where `p` denotes weight-shared cross-attention to the geometry branch and `s` denotes weight-shared split self-attention within surface and volume branches (with four surface blocks). The interaction radius is set to 0.25, with 65,536 geometry input points, 16,384 surface anchor points, and 16,384 geometry supernodes.

For the DrivAerML benchmark, we report results directly from the original work (Alkin et al., 2025). In this experiment, models were trained for 500 epochs with a batch size of 1 using the LION optimizer (peak learning rate $5 \times 10^{-5}$, weight decay $5 \times 10^{-2}$, 5% linear warm-up followed by cosine decay to $1 \times 10^{-6}$), float16 mixed-precision training, and a mean squared error loss. Training required approximately 7 hours for the DrivAerML setting, respectively, on a single NVIDIA H100 GPU with a memory footprint of roughly 4 GB.

**Transolver.** Transolver (Wu et al., 2024b) is a Transformer-based surrogate model that introduces the *Physics-Attention* mechanism for efficient learning on irregular geometries. At the time of writing, it represents the state-of-the-art on the ShapeNet-Car benchmark. Instead of applying self-attention directly to all mesh points, Transolver maps the input point cloud to a compact set of learnable *slices* (also referred to as physics tokens), where points with similar physical characteristics are softly assigned to the same slice. Specifically, each mesh point is first mapped to a set of slice weights, indicating its degree of association with each slice. These weights are then used to aggregate point-wise features into physics-aware tokens. Multi-head self-attention is applied at the slice level, significantly reducing the computational and memory costs compared to point-wise attention. Finally, the updated slice representations are projected back to the original mesh through a deslicing operation, followed by a point-wise feed-forward network.

In our experiments, we implement Transolver following the official implementation provided in (Wu et al., 2024b). For the NASA-CRM dataset, due to the original Transolver's memory bottlenecks, we employ a layer number of 8 with channels set as 256. For AhmedML and DrivAerML, as geometry amortized training is applied, we employ a subset size of 100k to support larger model size. On these two benchmarks, Transolver is set to have 16 layers and 256 channels.

**Transolver++.** Transolver++ (Luo et al., 2025) extends the original Transolver framework by scaling the Physics-Attention mechanism to million-scale meshes through multi-GPU distributed parallelism, while preserving the core architectural components of Transolver. By distributing both memory and computation across multiple GPUs, Transolver++ overcomes the memory bottleneck of the original Transolver (which is typically limited to meshes of approximately $7 \times 10^5$ cells), enabling training and inference on substantially larger unstructured meshes without altering the underlying attention formulation. The Physics-Attention mechanism remains central, aggregating irregular mesh information into a compact set of physics tokens for efficient self-attention, followed by deslicing back to the full-resolution mesh.

In our experiments, we configure Transolver++ to share the same model configurations as Transolver-3 to ensure a fair comparison. That is, on NASA-CRM and DrivAerML, Transolver++ has 24 layers with 256 channels. on AhmedML, the layer number is set to be 16.

### A.4. Numerical Stability

During physical state caching, the physical states $\mathbf{s}_j$ are calculated as $\mathbf{s}_j = \frac{\sum_{i=1}^{N} \mathbf{w}_{ij} \mathbf{x}_i}{\sum_{i=1}^{N} \mathbf{w}_{ij}}$. When $N$ becomes very large ($\sim 10^8$), there may exist overflow risks. As we use FP32 accumulation with maximum supported value at $3.4 \times 10^{38}$, the overflow risk is negligible. Regarding potential precision loss, our current implementation remains stable in all experiments. For absolute robustness, we can use FP64 calculation or adopt the Weighted Welford's Algorithm (Efanov et al., 2021).

## B. Standard Deviations

We repeat experiments three times on the NASA-CRM, AhmedML and DrivAerML datasets. The standard deviations are provided in Table 10. Note that we compare Transolver-3 with the second-best model, which is a strong baseline and is not achieved with a single model. The results demonstrates that Transolver-3 significantly outperforms baseline models, except in the case of volume pressure on AhmedML, showing the robustness of Transolver-3.

*Table 10.* Standard deviations on NASA-CRM, AhmedML and DrivAerML.

| MODELS | NASA-CRM | | AHMEDML | | | | DRIVAERML | | | |
|---|---|---|---|---|---|---|---|---|---|---|
| | $p_s$ | $C_f$ | $p_s$ | $u$ | $\tau$ | $p_v$ | $p_s$ | $u$ | $\tau$ | $p_v$ |
| SECOND-BEST MODEL* | 9.51 | 6.43 | 3.20 | 1.78 | 4.85 | **2.07** | 3.82 | 4.70 | 6.42 | 6.08 |
| **TRANSOLVER-3** STANDARD DEVIATION | **8.72** ±0.03 | **5.85** ±0.01 | **2.96** ±0.05 | **1.60** ±0.02 | **4.81** ±0.03 | 2.16 ±0.02 | **3.71** ±0.03 | **4.14** ±0.02 | **5.85** ±0.02 | **5.72** ±0.03 |

# C. More Visualizations

As a supplement to Figure 7 of the main text, in this section, we will provide detailed visualization of the physical states of Transolver-3 as well as case study showcases on different datasets.

## C.1. Physics States Visualization

For clarity, we visualize the physics-aware slice weights learned by Transolver-3 on the DrivAerML surface dataset.

Specifically, we present representative slice weights extracted from the first Physics-Attention layer, illustrating how Transolver-3 organizes surface points into structured physical states under complex automotive geometries. These visualizations highlight the model's ability to form coherent and spatially meaningful physics-aware tokens, reflecting its effectiveness in capturing salient aerodynamic patterns on high-resolution surface meshes.

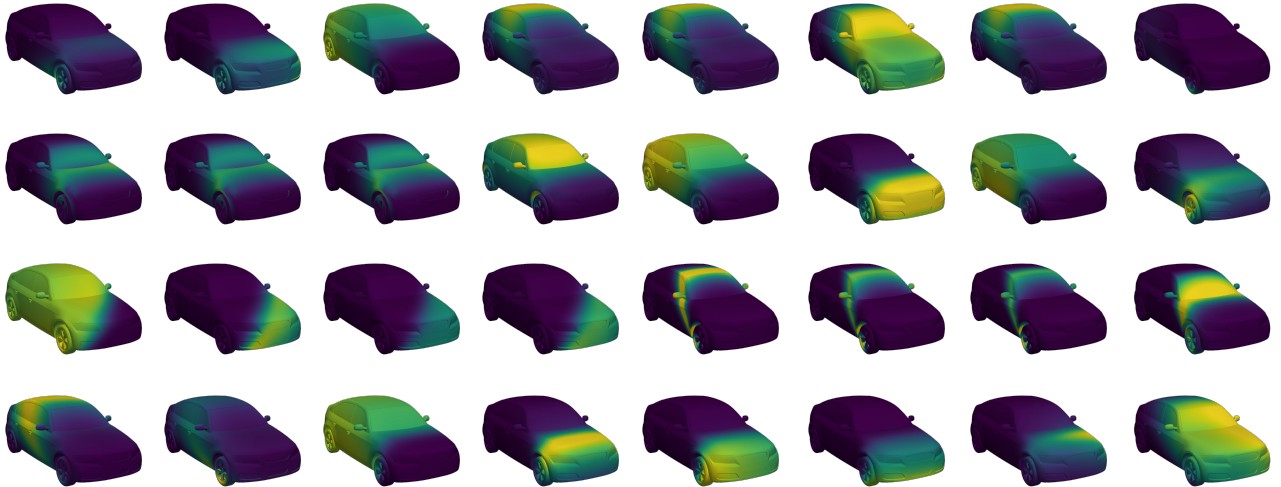

*Figure 10.* Visualizations of 32 physical states learned in the first layer of models on DrivAerML Surface. The lighter color means a higher weight in the corresponding physical state.

## C.2. Showcases

To better show the superior performance of Transolver-3 on three benchmarks of industrial-scale simulations, we provide additional showcases in Figure 11 – 15. These showcases include the predictions and error maps of Transolver-3 and two previous state-of-the-art baselines, Transolver++ and AB-UPT, on NASA-CRM, AhmedML, and DrivAerML, covering multiple physical quantities. As shown in these showcases, Transolver-3 is more effective in certain key and challenging geometric regions, such as the wings and tail of aircraft, the sharply varying rear regions of Ahmed car body geometry, and the windward regions of DrivAer geometry.

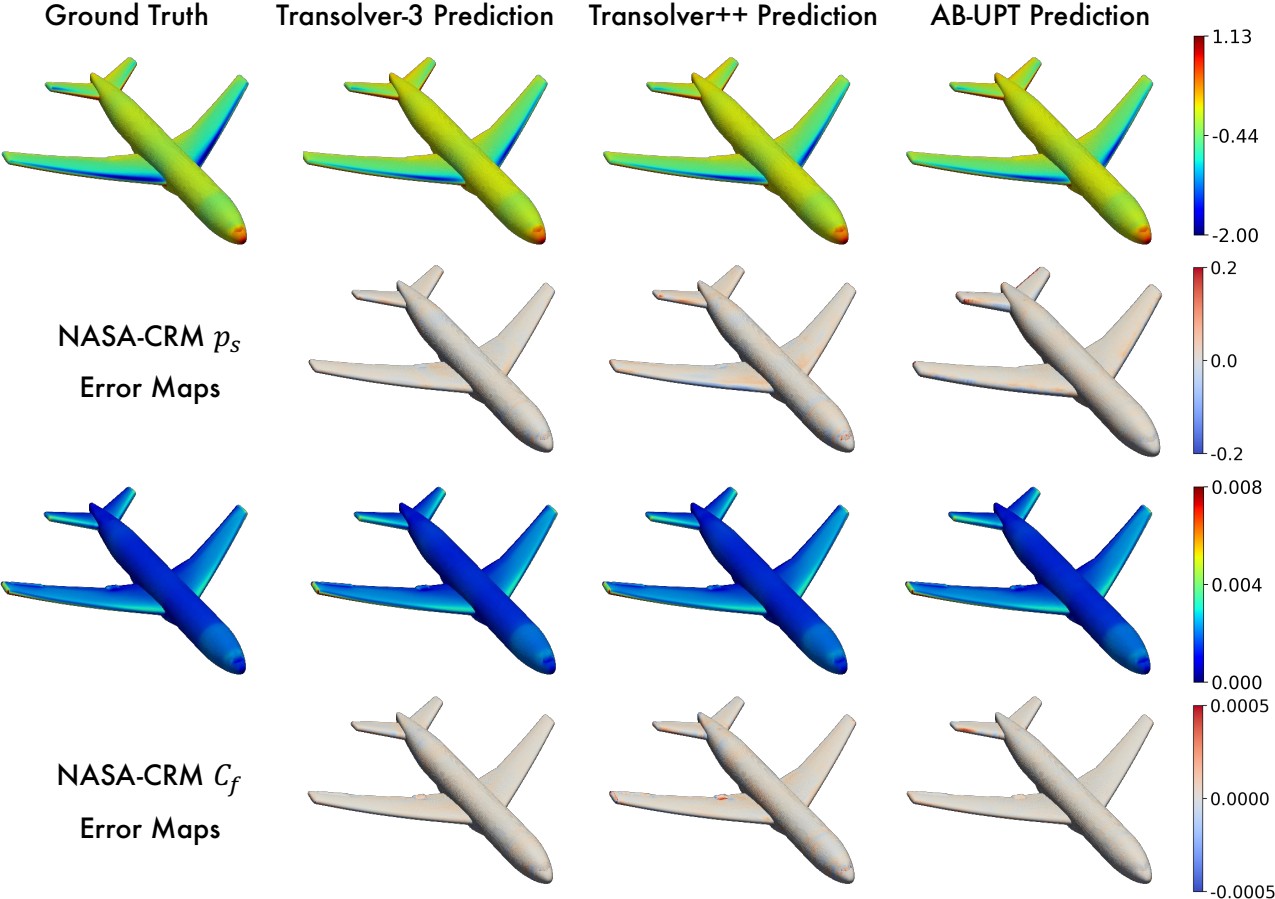

*Figure 11.* Showcases comparison with Transolver-3, Transolver++ and AB-UPT on NASA-CRM surface pressure $p_s$ and skin friction coefficient $C_f$. Lighter colors in the error maps correspond to lower prediction errors.

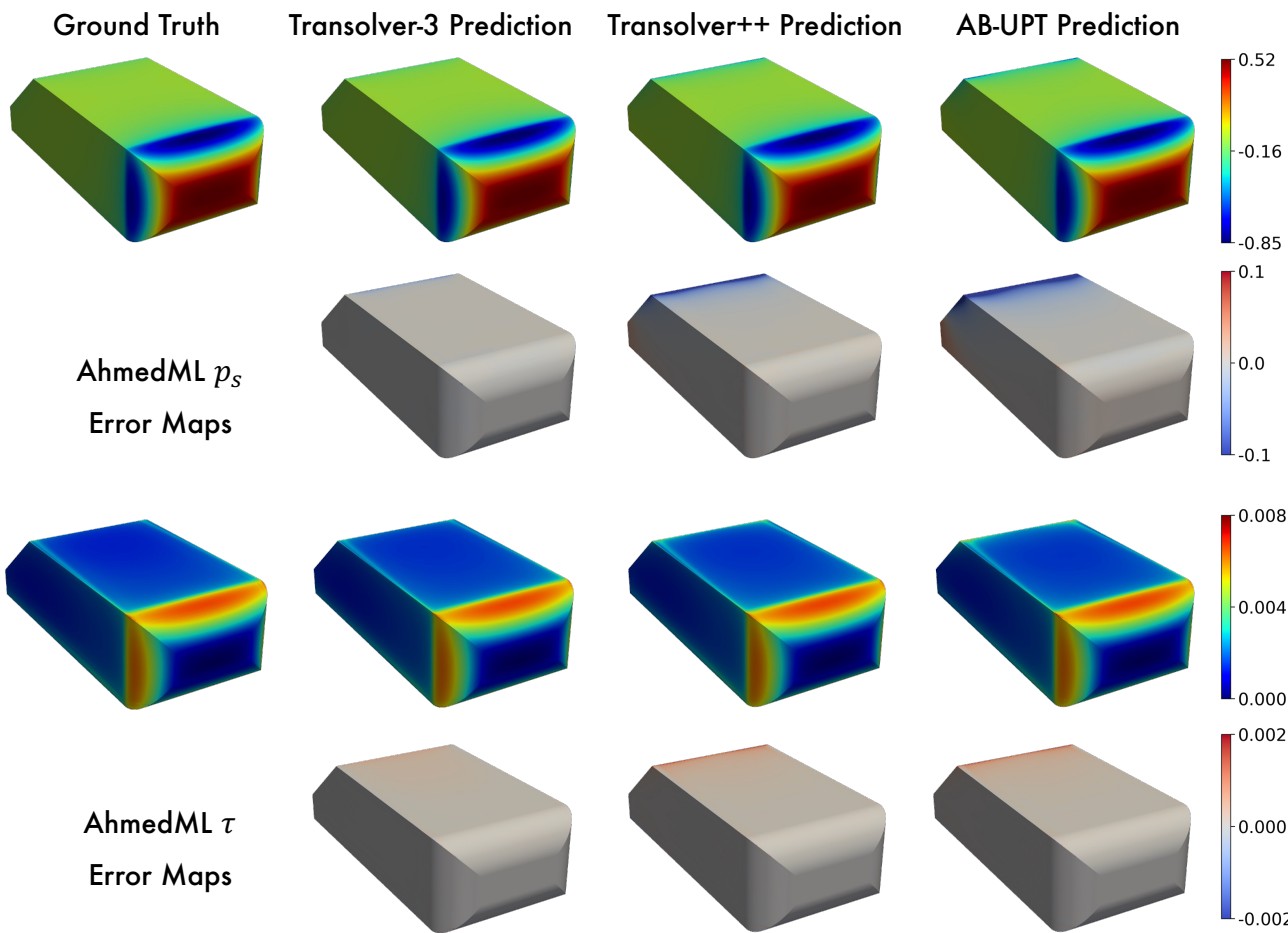

*Figure 12.* Showcases comparison with Transolver-3, Transolver++ and AB-UPT on AhmedML surface pressure $p_s$ and wall shear stress $\tau$. Lighter colors in the error maps correspond to lower prediction errors.

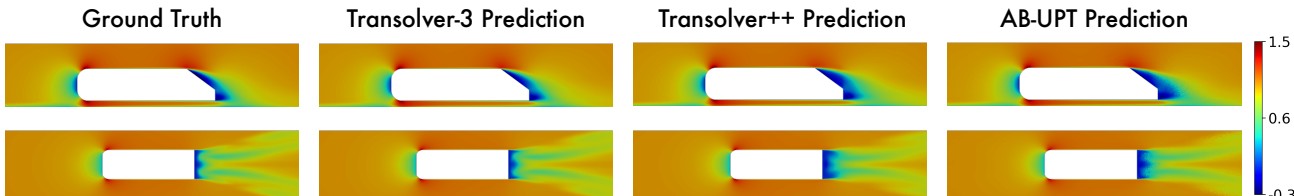

*Figure 13.* Showcases comparison with Transolver-3, Transolver++ and AB-UPT on AhmedML x-component of volume velocity $u$. The figures in the first and second rows represent slices from the side view and the bottom view, respectively.

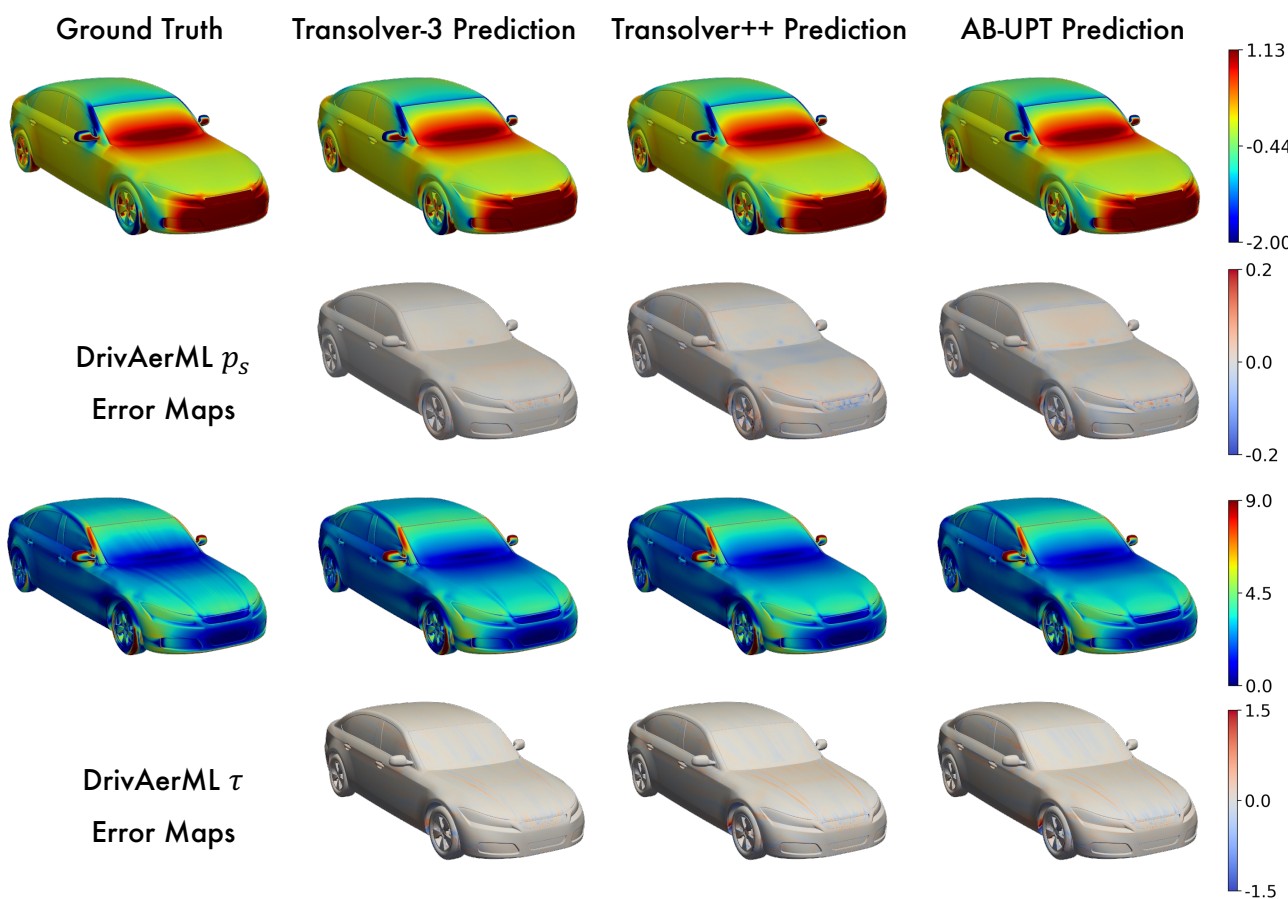

*Figure 14.* Showcases comparison with Transolver-3, Transolver++ and AB-UPT on DrivAerML surface pressure $\boldsymbol{p}_s$ and wall shear stress $\boldsymbol{\tau}$. Lighter colors in the error maps correspond to lower prediction errors.

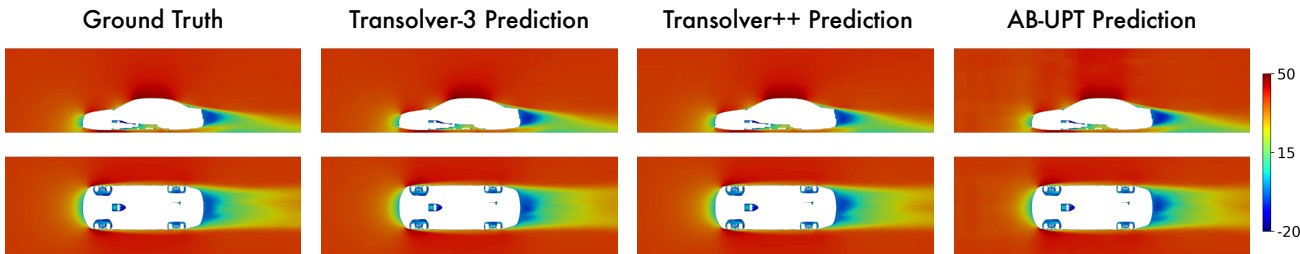

*Figure 15.* Showcases comparison with Transolver-3, Transolver++ and AB-UPT on DrivAerML x-component of volume velocity $\boldsymbol{u}$. The figures in the first and second rows represent slices from the side view and the bottom view, respectively.

# D. Limitations and Future Work

While Transolver-3 successfully scales neural PDE solvers to industrial-grade geometries exceeding $10^8$ cells, several limitations remain to be addressed. First, the geometry slice tiling mechanism achieves memory efficiency by trading computation for spatial capacity, meaning that the extra computation overhead requires practitioners to carefully choose the tile size to balance processing time against hardware constraints. Second, our framework has primarily been validated on stationary aerodynamic benchmarks; its generalization to highly transient or tightly coupled multi-physics systems has yet to be fully explored. Moving forward, we envision leveraging Transolver-3 as a scalable backbone for grand-scale physics foundation models by integrating structural geometry pre-training frameworks like GeoPT (Wu et al., 2026) to unlock robust cross-domain generalization across industrial topologies.

