# OpenReview forum: "Transolver-3: Scaling Up Transformer Solvers to Industrial-Scale Geometries"
_ICML.cc/2026/Conference — ICML 2026 regular_

### Official Review · Reviewer_LGZT · 2026-03-03

**Soundness:** 3
**Presentation:** 3
**Significance:** 2
**Originality:** 2
**Overall Recommendation:** 4
**Confidence:** 3

**Summary:**

Transolver-3 addresses the prohibitive memory and computational costs of processing high-resolution meshes in neural PDE solvers. It introduces two structural optimizations, faster slice/deslice and geometry slice tiling, which exploit matrix associativity and sequential computation to eliminate massive intermediate tensors. To handle meshes that still exceed GPU capacity, the authors propose a geometry amortized training and a decoupled inference framework that caches physical states for high-fidelity global predictions. The model handles up to 160 million cells, outperforming state-of-the-art baselines on the benchmarks.

**Compliance With Llm Reviewing Policy:**

Affirmed.

**Final Justification:**

The authors present a technically sound framework for scaling neural PDE solvers to industrial-scale geometries. The core contributions, faster slice/deslice, geometry slice tiling, and decoupled inference, are well-motivated and clearly validated. The rebuttal addressed most of my concerns. Q2 is theoretically grounded through the unbiased estimator argument, but the high-variance behavior under irregular fields remains empirically unverified. I support acceptance with minor revision, with the expectation that this limitation is explicitly discussed in the final version.

**Key Questions For Authors:**

- **Generalization Across PDE Families.** Have you evaluated whether geometry amortized training generalizes equally well to PDE families beyond incompressible CFD? While the proposed framework demonstrates strong scalability on aerodynamic benchmarks, it remains unclear whether similar performance and stability would hold in more complex PDE settings.
- **Training Efficiency and Convergence.** Although the paper reports FLOPs and inference latency comparisons with Transolver, it would be important to also compare end-to-end training time until converge against existing methods such as Transolver and Transolver++.

**Limitations:**

See the weakness and questions.

**Strengths And Weaknesses:**

**Strengths**
- The model scales to an industry-level applications that previously challenge to existing neural PDE solvers.
- The optimizations for faster slice and deslice are mathematically equivalents to the original process but significantly reduce memory and time cost.
- The introduction of geometry slice tiling and state caching provides a novel way to bypass the hardware limitations of single-GPU memory.

**Weaknesses**
- **Memory-time Tradeoff.** Geometry slice tiling saves memory usage but scarifies computational efficiency. smaller tile sizes increase training latency, and the tradeoff between memory and speed must be carefully tuned. Additionally, the multi-stage pipeline (tilling, amortized training, and decoupled inference) increase implementation complexity compared to simpler end-to-end schemes.
- **Assumptions behind Geometry Amortized Training.** Geometry amortized training randomly downsamples mesh cells during training. However, in the presented datasets (as observed in Figure 10-15), the physical fields appear relatively smooth that doesn’t seems not require extremely dense meshes (e.g., 10^8 cells) to represent them. In such cases, the necessity of ultra high-resolution meshes may partly stem from simulation conventions rather than intrinsic physical complexity. Also, it remains unclear how the proposed framework performs on more highly fluctuating or turbulent fields where extremely fine meshes are genuinely required to resolve sharp gradients or multi-scale structures.
- **Generality across PDE types.** The validation focuses primarily on incompressible aerodynamic CFD benchmarks under similar flow settings (e.g., wind-driven external flows, as illustrated in Figure 4). It is unclear whether similar scalability benefits would extend to other PDE families, especially those involving discontinuities, shocks, or Multiphysics coupling.

---

> ### Author Rebuttal · Authors · 2026-03-31
>
> We sincerely thank Reviewer LGZT for providing valuable feedback and suggestions.
>
> > **Q1:** "Geometry slice tiling saves memory usage but scariﬁes computational efficiency." "The multi-stage pipeline increases implementation complexity compared to simpler end-to-end schemes."
>
> We thank the reviewer for the comments.
>
> **On the memory-time trade-off:** We have provided a quantitative analysis of this trade-off in $\underline{\text{Table 5}}$. The results demonstrate that with an appropriate tile size (e.g., 100k), geometry slice tiling introduces only $6\%$ increase in training latency, but saves over $40\%$ of GPU memory. We believe this is a highly favorable trade-off which enables training of high-fidelity models on limited hardware.
>
> **About implementation complexity:** Our multi-stage training-inference pipeline is a necessary response to the scale of industrial problems. As illustrated in our introduction, training neural solvers end-to-end on geometries with over $10^8$ cells can be computationally prohibitive. Our pipeline comprising geometry amortized training and a decoupled inference strategy is specifically designed for computational tractability and enables Transolver-3 to handle unprecedented scales without sacrificing inference fidelity.
>
> > **Q2:** "The necessity of ultra high-resolution meshes may partly stem from simulation conventions rather than intrinsic physical complexity."
>
> We thank the reviewer for the insightful comment. The adoption of $10^8$​​ mesh cells in the DrivAerML dataset is a choice driven by physical fidelity rather than mere simulation convention. As the authors of DrivAerML emphasize, traditional RANS methods often yield unreliable correlations with experimental measurements in complex flows. In contrast, the Hybrid RANS-LES (HRLES) approach utilized in the dataset achieves greater accuracy by directly "resolving some of the turbulent motion." Therefore, we assume that the high-resolution discretization in this dataset is indispensable for meeting stringent engineering accuracy requirements.
>
> Meanwhile, our mesh scaling experiments in $\underline{\text{Figure 9 (a)}}$ provide direct empirical evidence that higher resolution is not redundant. We observe that the prediction error consistently decreases as the number of input mesh cells increases, even far beyond the training subset size. By scaling the input resolution, Transolver-3 effectively reduces discretization error and achieves higher-fidelity predictions.
>
> Moreover, as established in our response to **Reviewer SGGf Q1**, our geometry amortized training provides an unbiased estimator of the full-mesh gradient. Amortized training and decoupled inference ensure that our model actually utilizes the full mesh information both during training and inference. Our strategy does not discard information, but rather provides a mathematically rigorous and computationally efficient path to optimize the model over the entire mesh.
>
> > **Q3:** "How the proposed framework performs on more highly ﬂuctuating or turbulent ﬁelds" "It is unclear whether similar scalability beneﬁts would extend to other PDE families, especially those involving discontinuities, shocks, or Multiphysics coupling."
>
> Thank you for your questions. We clarify that Transolver-3 is fully capable of handling  more complex, unsteady, and multi-physics scenarios. To demonstrate generalizability beyond steady-state aerodynamics, we extend our experiments to REALM (Mao et al., 2025) benchmark.
>
> We compare Transolver-3 with AB-UPT on the FacadeFire dataset, which is a challenging **time-dependent, multi-physics** scenario involving coupled fluid-thermal dynamics. The relative L2 error across 20 prediction steps is shown below. Transolver-3 outperforms AB-UPT on this challenging dataset, demonstrating its generalizability to  unsteady multi-physics problems.
>
> | **Method**| Result   |
> |-|-|
> | AB-UPT| 30.6 |
> | **Transolver-3** | **18.7** |
>
> > **Q4:** "it would be important to also compare end-to-end training time until converge against existing methods such as Transolver and Transolver++."
>
> Thank you for your question on training time. To evaluate the training efficiency, we compare the total GPU hours required for convergence on the DrivAerML dataset across different architectures. The results are summarized below.
> |Method|Training Time/GPU Hours|
> |-|-|
> |Transolver|32|
> |Transolver++|45|
> |Transolver-3|36|
> |AB-UPT|40|
>
> Note that Transolver++ and Transolver-3 use gradient checkpointing to reduce memory usage, while Transolver does not. Transolver-3 is significantly faster than Transolver++ due to faster slice and deslice operations. Considering the superior performance of Transolver-3 over Transolver, the increased training time is acceptable.

---

> > ### Author Rebuttal · Reviewer_LGZT · 2026-04-04
> >
> > Thank you for the detailed rebuttal. Several of my concerns have been partially addressed, but I have remaining follow-up questions for the authors.
> >
> > On Q2 (Necessity of ultra high-resolution meshes): My original concern was whether the framework generalizes to fields with sharp gradients or multi-scale turbulent structures, not just to smooth aerodynamic flows. The DrivAerML dataset, while high-resolution, still represents relatively smooth flow fields. Could the authors clarify or provide evidence of how geometry amortized training performs when the underlying physical field is highly irregular or discontinuous?
> >
> > On Q3 (Generalization across PDE families): I appreciate the additional experiment on the FacadeFire dataset from REALM. However, this single benchmark involves coupled fluid-thermal dynamics and is time-dependent. My concern about generalization to PDEs involving discontinuities, shocks, or hyperbolic systems remains unaddressed. Could the authors comment on whether there are fundamental architectural barriers to handling such cases, or provide any preliminary evidence?

---

> > > ### Author Response · Authors · 2026-04-06
> > >
> > > Dear Reviewer LGZT:
> > >
> > > Thank you very much for your response to our rebuttal. We are glad that we have resolved some of your concerns. Below are our responses to your remaining follow-up questions.
> > >
> > > > On Q2: "Could the authors clarify or provide evidence of how geometry amortized training performs when the underlying physical field is highly irregular or discontinuous?"
> > >
> > > Thank you for your question. As discussed in our response to **Reviewer SGGf Q1**, the gradient derived from sampled subsets is an unbiased estimator of the full-mesh gradient. This property holds regardless of the field's smoothness. Moreover, in industrial simulations, cell density is typically higher around physics-critical regions (e.g., shock fronts or boundary layers). By performing uniform sampling in the index space, Transolver-3 naturally priortizes high-gradient regions during training. This effectively concentrates the optimization effort on resolving sharp discontinuities and irregular structures.
> > >
> > > > On Q3: "My concern about generalization to PDEs involving discontinuities, shocks, or hyperbolic systems remains unaddressed."
> > >
> > > Thank you for your question. To evaluate the performance of Transolver-3 on PDEs involving discontinuities, shocks, or hyperbolic systems, we conduct experiments on the AirCraft dataset. This dataset is proposed by Transolver++ and involves transonic and supersonic flows with clear shocks and discontinuities. The results comparison with AB-UPT is shown below (Relative L2 error in %).
> > >
> > > | Variable         | $p$      |
> > > | ---------------- | -------- |
> > > | AB-UPT           | 6.93     |
> > > | Transolver++     | 6.40     |
> > > | **Transolver-3** | **6.07** |
> > >
> > > Note that the AirCraft dataset contains around ~300k points per sample, and we employ a subset size of 100k when training Transolver-3. Transolver-3 outperforms both Transolver++ and AB-UPT on this challenging dataset, demonstrating that there are no architectural barriers to Transolver-3 handling systems involving discontinuities or shocks.
> > >
> > > Thank you again for your time and dedication.

---

### Official Review · Reviewer_vBj1 · 2026-03-11

**Soundness:** 2
**Presentation:** 3
**Significance:** 3
**Originality:** 2
**Overall Recommendation:** 5
**Confidence:** 3

**Summary:**

This paper addresses the challenge of scaling neural PDE solvers to industrial-scale geometries with over $10^8$ cells, where memory and latency bottlenecks make prior transformer solvers difficult to deploy. The main contribution is the development of Transolver-3, which introduces a systems-level redesign of the Physics-Attention mechanism. The proposed method incorporates faster slice and deslice operations, geometry slice tiling, geometry amortized training, and decoupled inference. Table 3 reports scaling up to the DrivAerML volume dataset with ~160M cells. Table 4 reports strong Relative L2 errors, with the paper claiming state-of-the-art performance across three datasets compared to other baselines.

**Compliance With Llm Reviewing Policy:**

Affirmed.

**Final Justification:**

My concerns about technical details have been resolved. But the concern about the limitation of novelty is not fully eliminated as the techniques are well-integrated but conceptually similar to other works. Given the strengthened empirical evidence and the practical significance of scaling to industry-level geometries, I raise my score.

**Key Questions For Authors:**

1. Could you provide an end-to-end training efficiency comparison (total wall-clock time, GPU hours) against other baselines (e.g., AB-UPT)?
2. Can you clarify the exact sampling strategy used in Geometry Amortized Training? Is it uniform random sampling, or does it account for the non-uniform density of the original mesh (e.g., boundary layers and wakes)?
3. Could you provide an ablation study on accuracy/efficiency across all the proposed methods (faster slice/deslice operations, geometry slice tiling, and geometry amortized training)? This would demonstrate which component contributes the most to memory savings and performance improvements.
4. How does performance degrade as the amortized training subset size decreases, and what is the minimum subset regime before accuracy collapses?

**Limitations:**

As detailed all above would be the limiation from my prespective, and please add a dedicated limitations section in the main paper. If all my concerns have been solved, I would increase the score.

**Strengths And Weaknesses:**

**Strengths:**

- The scaling bottlenecks are carefully analyzed and addressed with concrete, mathematically sound algorithmic and system changes. The experimental scope is broad and includes very large meshes with up to 160M cells.
- The problem has high practical importance for AI-CAE and industrial surrogate modeling. Demonstrated scalability to the $10^8$-cell scale is highly impactful and bridges a critical gap between academic research and real-world engineering workflows.
- The architecture and complexity arguments are generally clear, and the benchmark scale is compelling. The ablation studies on memory footprint and latency trade-offs are informative.

**Weaknesses:**
- Part of the advantage comes from engineering optimization and training/inference strategy design. Fairness of baseline reproduction under identical optimization effort is difficult to fully verify. As noted in the Table 4 footnote, baselines marked with an asterisk (Graph U-Net, GINO, GAOT, Transolver, Transolver++) could not natively handle the large meshes on AhmedML and DrivAerML, so the authors applied geometry amortized training to them and, during evaluation, split the input mesh into independent chunks whose outputs were simply concatenated. This chunk-wise independent inference is fundamentally inferior to Transolver-3's decoupled inference with global physical state caching, which introduces a systematic advantage that is not purely attributable to the model itself. Intuitively, the major reduction in memory stems from the geometry amortized training. Thus, the paper lacks a rigorous ablation study on accuracy and efficiency for its individual core components. While Figure 6 shows memory improvements, it is unclear how much the predictive accuracy degrades if Tiling is removed and the model is forced to use a smaller amortized subset.
- The paper lacks a dedicated limitations section in the main text. Additionally, the "sampling" strategy for Geometry Amortized Training is under-specified; it is unclear if this is uniform random sampling, if it accounts for the highly non-uniform density of industrial meshes, or whether there is a strategic mechanism to ensure adequate coverage without redundantly sampling the same regions.
- Since this is not the only work capable of handling such large-scale meshes, the impact claims would be stronger with more transparent resource accounting. The paper lacks an end-to-end training efficiency comparison (e.g., total wall-clock time or GPU hours to convergence) against other strong baselines, such as AB-UPT. Furthermore, the memory and latency comparisons (Figure 6, Figure 8) are only conducted against previous Transolver variants, omitting other models like AB-UPT entirely. Given that AB-UPT achieves very competitive accuracy and is also designed for large-scale meshes, a direct comparison of memory usage, training cost, and inference latency against AB-UPT is essential to substantiate the claimed efficiency advantages.
- The originality of the current work appears limited. Most contributions are incremental refinements within the Transolver family, and its core techniques are conceptually similar to prior ideas: tiling-based memory optimization (FlashAttention), random point/subset sampling (UPT, GAOT, AB-UPT), and truncated-domain or patch-based sampling (P3D). Overall, the novelty seems to lie more in existing system integration and engineering optimization than in fundamentally methodology.

---

> ### Author Rebuttal · Authors · 2026-03-31
>
> We sincerely thank Reviewer vBj1 for providing valuable feedback and suggestions.
>
> > **Q1:** "a systematic advantage that is not purely attributable to the model itself."
>
> Thank you for your comment. We clarify that the "systematic advantage" of physical state caching is an intrinsic advantage of our architecture. The additive nature of Physics-Attention allows for the decoupling of physical state computation from field inference, enabling Transolver-3 to represent global context through a compact $O(M)$ physical state tensor. For baseline models, global representation would require $O(N)$ memory, which is computationally prohibitive. Therefore, this system-level efficiency is also an algorithmic contribution of Transolver-3 design.
>
> > **Q2:** "an ablation study on accuracy/efficiency across all the proposed methods?"
>
> Thank you for this valuable question.
>
> Firstly, without geometry amortized training, it would be impossible for neural solvers to process industrial-scale meshes like DrivAerML. The ablation on subset size will be presented in the next question.
>
> Secondly, both faster slice/deslice operations and geometry slice tiling perform **equivalent transformations** on the original Physics-Attention, and do not influence the accuracy of the model.
>
> Therefore, we focus on efficiency analysis of the two optimizations. As for time complexity, $\underline{\text{Figure 8}}$ provides a thorough analysis of the efficiency gain of faster slice/deslice (note that geometry slice tiling is applied only during training and does not impact inference latency). $\underline{\text{Table 5}}$ also shows the memory-time trade-off of slice tiling, where we observe that an appropriate tile size induces little computational burden.
>
> As for memory savings, $\underline{\text{Figure 6}}$ also provides a clear analysis of faster slice/deslice and slice tiling. Note that "Transolver-3 w/o tiling" means Transolver-3 with faster slice/deslice and without slice tiling, which saves around 10% of memory compared to Transolver++. "Transolver-3 with tiling" is equipped with both tiling and faster slice/deslice, and it achieves a total 50% reduction. These results clearly show that geometry slice tiling contributes most to memory savings.
>
> > **Q3:** "How does performance degrade as the amortized training subset size decreases?"
>
> Thank you for your suggestions. We provide an ablation study of the training subset size on DrivAerML surface. The relative L2 error across various $n$ is shown below.
>
> |$n$|$p_s$|$\tau$|
> |-|-|-|
> |1k|4.84|7.31|
> |5k|4.05|6.39|
> |10k|3.86|6.22|
> |20k|3.80|6.08|
> |50k|3.76|6.02|
> |100k|3.71|5.85|
>
> The results reveal that model performance scales positively with $n$ yet remains robust even at $n=5k$. For the smallest $n=1k$, the accuracy degrades gracefully without collapsing. This demonstrates that geometry amortized training is an efficient and robust strategy enabling an optimal trade-off between efficiency and accuracy on industrial-scale geometries.
>
> > **Q4:** "The paper lacks a dedicated limitations section in the main text."
>
> Thank you for this constructive suggestion. We will add a detailed limitation section in the final version of our paper to discuss the limitations of current method and how they may be solved in the future. For example, we will discuss the performance trade-offs associated with the amortized subset size $n$, and unexplored methods for further efficiency gains via mixed-precision training..
>
> > **Q5:**  "The "sampling" strategy for Geometry Amortized Training is under-speciﬁed."
>
> Sorry for the ambiguity. Across all our experiments, we employ a **uniform sampling strategy over the cell indices**. In industrial CFD applications, computational meshes are inherently non-uniform, with significantly higher cell densities allocated to physics-critical regions. By sampling uniformly in the index space, our strategy naturally performs a form of implicit importance sampling and prioritizes high-resolution regions. This sampling strategy also ensures distribution consistency between training and inference.
>
> > **Q6:** "A direct comparison of memory usage, training cost, and inference latency against AB-UPT."
>
> Thank you for your valuable suggestions. Here we provide a detailed comparison of memory usage, training cost, and inference latency against AB-UPT on DrivAerML dataset.
>
> |Method|Training Memory/GB|Training Cost/GPU hours|Inference Memory/GB|Inference Latency/s|
> |-|-|-|-|-|
> |AB-UPT|10.0|40|11.2|220|
> |Transolver-3|12.5|36|6.0|260|
>
> Transolver-3 achieves comparable computational efficiency to AB-UPT. Despite slightly higher training memory, training speed is around 10% higher. The marginal increase in inference latency can also be justified by superior accuracy.
>
> > **Q7:** "the novelty seems to lie more in existing system integration and engineering optimization."
>
> We thank the reviewer for the suggestion to highlight our novelty. Please refer to **Reviewer SGGf Q6** for detailed discussion.

---

> > ### Author Rebuttal · Reviewer_vBj1 · 2026-04-03
> >
> > Thank you for the rebuttal and extended experiments. My concerns about technical details have been resolved. But the concern about the limitation of novelty is not fully eliminated as the techniques are well-integrated but conceptually similar to other works. Given the strengthened empirical evidence and the practical significance of scaling to industry-level geometries, I raise my score.

---

> > > ### Author Response · Authors · 2026-04-03
> > >
> > > Dear Reviewer vBj1,
> > >
> > > Thank you very much for your response and for raising the score. We are glad that our rebuttal and the extended experiments have addressed your technical concerns. We also sincerely appreciate your recognition of the strengthened empirical evidence and the practical significance of our work in scaling to industrial-level geometries. Regarding the conceptual novelty, we will further refine our discussion in the final version to better highlight the unique challenges we addressed in this work. We will also integrate all the newly added experiments into the revised paper.
> > >
> > > Thank you again for your time and dedication.

---

### Official Review · Reviewer_SGGf · 2026-03-12

**Soundness:** 3
**Presentation:** 4
**Significance:** 4
**Originality:** 3
**Overall Recommendation:** 5
**Confidence:** 4

**Summary:**

This paper proposes Transolver-3, a scalable neural PDE solver framework targeting industrial-scale meshes with over 10^8 cells. Building on the Transolver family’s Physics-Attention mechanism, the authors introduce two architectural optimizations (faster slice/deslice via matrix multiplication reordering and geometry slice tiling), a geometry amortized training strategy that trains on random mesh subsets, and a decoupled inference framework that caches physical states for full-mesh prediction. Experiments on three aerodynamic benchmarks (NASA-CRM, AhmedML, DrivAerML) with meshes up to 160M cells show state-of-the-art performance on 9 metrics and accurate prediction of drag and lift coefficients.

**Compliance With Llm Reviewing Policy:**

Affirmed.

**Final Justification:**

The rebuttal has adequately addressed my main concerns. I raise my score from 3 to 5.

**Key Questions For Authors:**

1. How sensitive is the model performance to the subset size $n$ in geometry amortized training? Could you provide an ablation study showing how training-phase subset size affects final prediction accuracy across all three benchmarks? If a critical threshold exists below which performance degrades significantly, this would be important for practitioners.

2. What is the total end-to-end inference wall-clock time on DrivAerML (160M cells), including both physical state caching and full mesh decoding? How does this compare with the original CFD simulation time? This information is essential for assessing practical industrial utility. A positive answer showing substantial speedup over CFD would strengthen the paper.

3. The physical state cache is accumulated from chunks across the full mesh. Have you observed any numerical stability issues (e.g., floating-point accumulation errors) when the number of chunks K becomes very large (e.g., for 160M cells with chunk sizes of ~400K, $K$ ≈ 400)? If numerical issues exist, how are they mitigated?

4. Can the decoupled inference framework be applied to time-dependent simulations where physical states evolve over time? Or is it fundamentally limited to steady-state prediction? Clarifying this would help scope the generalizability of the approach.

**Limitations:**

The authors provide a brief Impact Statement but lack a dedicated limitations discussion. Key unaddressed points include: generalizability beyond steady-state aerodynamics, small dataset sample counts, computational cost of training on multi-TB datasets, and potential failure modes of the amortized training strategy. A concise limitations paragraph would improve the paper. No significant ethical concerns are identified.

**Strengths And Weaknesses:**

**Strengths**
- The complexity analysis is thorough and convincing. Tables 1–2 clearly show how the optimized Physics-Attention reduces N-related operations, and the mathematical equivalence of the faster slice/deslice is rigorously established. The experimental evaluation covers three benchmarks of increasing scale with seven competitive baselines, and the memory/latency analyses (Figure 6, 8) provide solid empirical support for the claimed efficiency gains.
- The paper is well-organized with a clear narrative arc from complexity analysis to training-phase and inference-phase solutions. The figures are effective, and the pseudocode in the appendix aids reproducibility. The analogy to LLM scaling techniques is insightful and helps readers from the broader ML community appreciate the contributions.
- Scaling neural PDE solvers to 160M-cell industrial meshes addresses a genuine and important gap between academic research and industrial deployment. The strong R² values on drag/lift coefficient prediction are directly relevant to real-world automotive and aerospace design workflows, and the 1.9× single-GPU capacity gain and ~60% latency reduction are practically meaningful.
- Although individual techniques are inspired by known ideas, their systematic adaptation to the Physics-Attention framework is well-motivated. The insight that physical state computation is inherently decomposable across mesh chunks, enabling the decoupled caching-based inference, is a useful architectural contribution specific to the Transolver family.

**Weaknesses**
- The geometry amortized training strategy lacks theoretical justification. No convergence analysis or error bounds are provided for training on random subsets, and an ablation on the training subset size is missing. Additionally, statistical significance measures (e.g., error bars) are absent from Table 4, making it hard to judge whether some of the reported improvements are meaningful given the small test set sizes.
- The total end-to-end inference time on DrivAerML (physical state caching + full mesh decoding) is not explicitly reported. For the target audience of industrial practitioners, this is a critical number. A wall-clock comparison with traditional CFD solvers would also greatly strengthen the practical narrative.
- The evaluation is limited to steady-state aerodynamic simulations. The generalizability to time-dependent problems, other PDE families, or multi-physics scenarios is not discussed, which limits the demonstrated breadth of impact.
- The core techniques—matrix multiplication reordering, tiled computation, random subsampling, and caching—are well-established in the deep learning systems and point cloud communities. The novelty lies more in careful engineering integration than in fundamentally new algorithmic ideas. The paper could better articulate what challenges were unique to the Transolver setting.

---

> ### Author Rebuttal · Authors · 2026-03-31
>
> We sincerely thank Reviewer SGGf for providing valuable feedback and suggestions.
>
> > **Q1:** About theoretical justification and convergence analysis of geometry amortized training.
>
> Thank you for your question. Below we discuss amortized training's foundation and its convergence behavior.
>
> Let $\mathcal{M}$ the full mesh with $N$ cells. The global loss is:
> $$
> L_{full}(\theta)=\frac{1}{N}\sum_{i=1}^N l(f_{\theta}(x_i),y_i)
> $$
> In each step, a subset $S$ of $n$ cells is uniformly sampled, with $P(x_i\in S)=n/N$. The gradient is:
> $$
> \mathbf{g}(\theta)=\frac{1}{n}\sum_{x_j\in S}\nabla_{\theta} l(f_{\theta}(x_j),y_j)
> $$
> $\mathbf{g}(\theta)$ is an unbiased estimator of full-mesh gradient:
>
> $$
> \mathbb{E}[\mathbf{g}(\theta)] = \frac{1}{n} \sum_{i=1}^N P(x_i \in S) \nabla_\theta l(f_\theta(x_i), y_i) = \nabla_\theta L_{full}(\theta),
> $$
>
> The convergence property is similar to SGD with rate $O(1/\sqrt{T})$. Gardient variance $\sigma^2\propto 1/n$, explaining Transolver-3's robustness and accuracy gains as $n$ increases.
>
> > **Q2:** About ablation study on training subset size.
>
> We provide ablation of training subset size on DrivAerML. The relative L2 error is shown below.
>
> |$n$|$p_s$|$u$|$\tau$|$p_v$|
> |-|-|-|-|-|
> |1k|4.84|5.30|7.31|7.94|
> |5k|4.05|4.63|6.39|6.53|
> |10k|3.86|4.49|6.22|6.19|
> |20k|3.80|4.29|6.08|5.90|
> |50k|3.76|4.21|6.02|5.79|
> |100k|3.71|4.14|5.85|5.72|
>
> Performance scales positively with $n$ yet remains robust even at $n=5k$. The accuracy degrades gracefully without collapsing, validating the efficiency and robustness of amortized training.
>
> > **Q3:** About statistical signiﬁcance and dataset sample counts.
>
> We repeat experiments in Table 4 across 3 trials with different random seeds. We follow the train-test split of AB-UPT and use all available data.
>
> |Dataset|NASA-CRM|| AhmedML|||| DrivAerML||||
> |-|-|-|-|-|-|-|-|-|-|-|
> |Variables|$p_s$|$C_f$|$p_s$|$u$|$\tau$|$p_v$|$p_s$|$u$|$\tau$|$p_v$|
> |mean$\pm$std|$8.72\pm0.03$|$5.85\pm0.01$|$3.03\pm0.05$|$1.59\pm0.02$| $4.78\pm0.03$ |$2.15\pm0.02$|$3.69\pm0.03$|$4.16\pm0.02$|$5.84\pm0.02$| $5.72\pm0.03$ |
>
> In 9 out of 10 experiments, the improvements over AB-UPT exceed three standard deviations ($3\sigma$), confirming statistical significance of our results..
>
> > **Q4:** About inference time on DrivAerML and comparison with traditional solvers.
>
> The inference of Transolver-3 on DrivAerML takes about 5 minutes on an A100 GPU. The original CFD simulation takes about 40 hours on 1536 CPU cores.
>
> > **Q5:** About generalizability to time-dependent problems, other PDE families or multi-physics scenarios.
>
> To demonstrate generalizability beyond steady-state aerodynamics, we extend experiments to REALM (Mao et al., 2025) benchmark on FacadeFire dataset, which is a **time-dependent, multi-physics** scenario with **fluid-thermal dynamics**. The relative L2 error across 20 prediction steps is shown below.
>
> |**Method**|Result|
> |-|-|
> |AB-UPT|30.6|
> |**Transolver-3**|**18.7**|
>
> The superior performance of Transolver-3 demonstrates its generalizability to unsteady multi-physics problems. Note that the decoupled inference framework is compatible with time-dependent tasks by updating the physical states at each timestep.
>
> > **Q6:** About technical novelty and challenges unique to the Transolver setting.
>
> We thank the reviewer for the suggestion to articulate our novelty. In ML Systems community, designing architectures to enable previously "prohibitive" scales is recognized as a core contribution (e.g., FlashAttention, ZeRO). Our work resolves scaling challenges of Transolver with unique training strategies and architecture designs.
>
> Specifically, we discover the **decoupling property** of latent physical states from field inference. This insight enables Transolver-3 to process industrial-scale geometries under limited memory and inspires amortized training.
>
> Moreover, the architectural optimizations are derived from a systematic complexity analysis. We shift the computational-intensive projections from mesh domain to latent domain, and internalize the multi-GPU parallelism of Transolver++ into Physics-Attention via tiling, which drastically boosts single-GPU capacity.
>
> To conclude, Transolver-3 is the first successful scaling of Transolver to industrial geometries, achieved with the fundamental discovery of decoupling property as well as principled ML system designs.
>
> > **Q7:** About numerical stability during physical state caching.
>
> During physical state caching, the physical states $s_j$ are calculated as $s_j=\frac{\sum_{i=1}^N w_{ij}x_i}{\sum_{i=1}^N w_{ij}}$. When $N$ becomes very large($\sim10^8$), there may exist overflow risks. However, as we use FP32 accumulation with maximum supported value at $3.4\times10^{38}$, the overflow risk is negligible. Regarding potential precision loss, our current implementation remains stable in all experiments. For absolute robustness, we can use FP64 calculation or adopt the Weighted Welford's Algorithm.

---

> > ### Author Rebuttal · Reviewer_SGGf · 2026-04-02
> >
> > The rebuttal solved my concerns. I have rised the score to 5.

---

> > > ### Author Response · Authors · 2026-04-02
> > >
> > > Dear Reviewer SGGf,
> > >
> > > Thank you very much for recognizing our rebuttal and for raising your score. We sincerely appreciate your constructive and detailed review, particularly your suggestions on theoretical justification, statistical analysis, and the important ablation study on subset size. We will integrate all the newly added experiments and theoretical clarifications into the revised paper.
> > >
> > > Thank you again for your time and dedication.

---

### Official Review · Reviewer_wQyX · 2026-03-12

**Soundness:** 3
**Presentation:** 4
**Significance:** 4
**Originality:** 3
**Overall Recommendation:** 5
**Confidence:** 3

**Summary:**

The authors present an extension to the Transolver unstructured PDE neural surrogate family that works on meshes with O(10^8) degrees of freedom. This is achieved by (1) re-ordering the operations for slicing and de-slicing degrees of freedom, (2) working on tiles of the mesh sequentially, and (3) training only on a subset of mesh nodes that allow for full usage of the GPU memory. During inference, the model operates on chunks and the intermediate activations are cached and re-used. The authors demonstrate state-of-the-art accuracy on large-scale engineering CFD datasets.

**Compliance With Llm Reviewing Policy:**

Affirmed.

**Final Justification:**

The authors have fully addressed all my concerns in the rebuttal. I apologize for the confusion that Transolver-3 would not allow predicting on arbitrary query points (like additional surface nodes).
I have raised my score to 5.

**Key Questions For Authors:**

1. Can the authors comment on weaknesses 1. and 2.?
2. Why do we have to work on high-resolution meshes in the first place? What would happen if we used downsampled high-fidelity solutions? The authors argue with the quadrature error in eq. 6. I am wondering if it is really quadrature error, if the error in volumetric fields and the surface fields is still in the O(2%) regime. Could the authors comment on the setup in figure 9 (b)?
3. Why is the channel count higher than the number of physical states $C \ge M$ (line 144, right column)?
4. Could the authors elaborate on how their architecture would compare against a hierarchical MeshGraphNet baseline (e.g., using a bi-strided coarsening mechanism)?

**Limitations:**

No, but I also do not see any major limitations besides a lack of compute/memory comparison with AB-UPT. Still, I would encourage the authors to add a section on relevant limitations.

**Strengths And Weaknesses:**

This piece of research introduces many clever engineering optimizations that enable Transolver-style surrogates to scale to very large meshes. The manuscript is well written, with polished figures and explanations. I found the reasoning behind their improvements clear, and empirical evidence underlines the efficacy. The quantitative results present (mostly significant) improvements over the next strongest baseline.

Weaknesses:
1. Since the authors did not provide any standard deviations or confidence intervals, it is hard to judge whether the (oftentimes small, though consistent) improvements over the next-best non-Transolver architecture (which in all scenarios is AB-UPT) listed in Table 4 are actually statistically significant.
2. Given that those improvements are significant, it would be important to include AB-UPT in a similar evaluation of compute cost and memory usage as is done for all three generations of Transolver models (figures 6 & 8, as well as table 5). Otherwise, it leaves me suspicious that AB-UPT could be the superior choice in a Pareto-optimal analysis.
3. The axes of figure 8 should be doubly-logarithmic to read off the scaling behavior.
4. It is unclear to me why the authors chose a basis-four logarithmic plot?
5. In eq. 1, the authors write $\mathbf{x}_{\text{proj}} = \text{Linaer1}(\mathbf{x}) = \mathbf{w}_{\text{Linear1}} \mathbf{x}$ which is inconsistent with the figure 2. The version in figure 2 should be correct to mix the channels.

Overall, I find this a great piece of research that deserves to be presented at ICML. I am willing to increase my score if the authors can elaborate on the compute/memory-comparison with AB-UPT.

---

> ### Author Rebuttal · Authors · 2026-03-31
>
> We sincerely thank Reviewer wQyX for providing valuable feedback and suggestions.
>
> > **Q1:** "The authors did not provide any standard deviations or conﬁdence intervals."
>
> Thank you for your suggestion. To ensure statistical significance, we repeat experiments of Transolver-3 in $\underline{\text{Table 4}}$ across 3 trials with different random seeds. As shown below, across 9 out of the 10 experiments, the improvements over AB-UPT exceed three standard deviations ($3\sigma$). The small standard deviations demonstrate that Transolver-3 is robust and stable.
>
> |Datasets|NASA-CRM|| AhmedML|||| DrivAerML||||
> |-|-|-|-|-|-|-|-|-|-|-|
> |Variables|$p_s$|$C_f$|$p_s$|$u$|$\tau$|$p_v$|$p_s$|$u$|$\tau$|$p_v$|
> |mean$\pm$std|$8.72\pm0.03$|$5.85\pm0.01$|$2.98\pm0.05$|$1.59\pm0.02$| $4.78\pm0.03$ |$2.15\pm0.02$|$3.69\pm0.03$|$4.16\pm0.02$|$5.84\pm0.02$| $5.72\pm0.03$ |
>
> > **Q2:** "Include AB-UPT in a similar evaluation of compute cost and memory usage."
>
> As requested, we provide the training and inference cost comparison on the DrivAerML dataset.
>
> |Method|Training Memory/GB|Training Cost/GPU hours|Inference Memory/GB|Inference Latency/s|
> |-|-|-|-|-|
> |AB-UPT|10.0|40|11.2|220|
> |Transolver-3|12.5|36|6.0|260|
>
> Transolver-3 achieves comparable computational efficiency to AB-UPT. Despite slightly higher training memory, training speed is around 10% higher. The marginal increase in inference latency can also be justified by superior accuracy.
>
> > **Q3:** "The axes of ﬁgure 8 should be doubly-logarithmic."
>
> We agree that a log-log plot is more appropriate for visualizing scaling behavior and will update it accordingly in our final version.
>
> > **Q4:** "Why the authors chose a basis-four logarithmic plot."
>
> Sorry for lack of clarity. To clarify, the data points were sampled at a basis-2 increment (10K, 20K,..., 2.56M). The x-axis labels were presented at basis-4 to prevent label overlap and enhance visual readability. To provide a more standard analysis, we will update $\underline{\text{Figure 8}}$ with basis-2 scaling.
>
> > **Q5:** "The version in ﬁgure 2 should be correct to mix the channels."
>
> We thank the reviewer for this question. While $\underline{\text{Figure 2}}$ ($xW_{\text{Linear1}}$) correctly reflects our implementation, Equation 1 contains a typographical error, and we will correct it in the final version as:
>
> $$x_{\text{proj}}=\text{Linear1}(x)=xW_{\text{Linear1}}.$$
>
> > **Q6:** "What would happen if we used downsampled high-ﬁdelity solutions?"
>
> Thank you for your question. Geometry amortized training allows training with random subsets from the full mesh. To verify the effect of training on a fixed subset, we conduct experiments on DrivAerML volume with a single subset containing 400k points. The results are shown below.
>
> |Variables|$u$|$p_v$|
> |-|-|-|
> |Transolver-3 (single subset)|4.37|6.09|
> |Transolver-3 (full subset)|**4.14**|**5.72**|
>
> While Transolver-3 can learn from a fixed subset, training on the full mesh yields superior accuracy, demonstrating the necessity of working on the high-resolution mesh.
>
> > **Q7:** "if the error in volumetric ﬁelds and the surface ﬁelds is still in the O(2%) regime." "the setup in ﬁgure 9 (b)?"
>
> Thank you for this observation. $\underline{\text{Figure 9 (b)}}$ investigates the sensitivity of global integral coefficients ($C_d$ and $C_l$) to the evaluation resolution. When the number of cells is small ($10^4$ cells), the errors in $C_d$ and $C_l$ are high, far exceeding the error achieved at full resolution. This demonstrates that while the prediction error is still in the O(2%) regime, the quadrature error dominates, confirming that full-mesh prediction is indispensable for obtaining engineering-grade accuracy.
>
> > **Q8:**  "Why is the channel count higher than the number of physical states?"
>
> We thank the reviewer for this observation. The configuration $C\geq M$ is motivated by empirical findings. We observe that $M=32$ or $64$ provide a sufficient bottleneck for global geometric representations, consistent with Transolver and Transolver++. The channel count ($C=128$ or $256$) is configured higher to ensure enough model capacity. While $C$ and $M$ remain within a comparable order of magnitude, we agree that the original notation may be slightly misleading. We will update the notation in the final version to better reflect their relationship.
>
> >  **Q9:**  "compare against a hierarchical MeshGraphNet baseline"
>
> Thank you for your suggestion. We conduct experiments with MeshGraphNet on all benchmarks. The relative L2 error is shown below.
>
> |Datasets|NASA-CRM||AhmedML||||DrivAerML||||
> |-|-|-|-|-|-|-|-|-|-|-|
> |Variables|$p_s$|$C_f$|$p_s$|$u$|$\tau$|$p_v$|$p_s$|$u$|$\tau$|$p_v$|
> |MeshGraphNet|17.4|11.6|4.98|4.34|5.58|5.09|14.8|16.3|21.4|23.2|
>
> Transolver-3 outperforms MeshGraphNet on all benchmarks, which we believe can be attributed to its global representation learning capability and scalability to industrial-scale meshes.

---

> > ### Author Rebuttal · Reviewer_wQyX · 2026-04-03
> >
> > I thank the authors for the extensive reply to my rebuttal, and I appreciate the additional experimental results. The reported standard deviations reinforce that the improvements over AB-UPT are statistically significant. I also thank the authors for promising to improve their manuscript with respect to Q3, Q4, Q5, and Q8.
> >
> > Below, I have a few comments on the rebuttal and some minor follow-up questions:
> >
> > > Re Q2:
> >
> > Thank you for providing these numbers. It is reassuring that Transolver-3 is broadly comparable to AB-UPT in training cost, memory usage, and inference latency. From a practitioner's perspective, both models can be reasonably inferenced on a workstation-grade GPU (40GB+ of VRAM).
> >
> > That said, I would like to note a few considerations. First, AB-UPT offers the additional capability of being a conditional field (for arbitrary query points), which Transolver-3 is not (which could play into my comments on Q6 & Q7). Second, AB-UPT explicitly encodes geometric priors through separate branches for surface and volume, which should be particularly beneficial for derived integral quantities. Third, the accuracy gap between Transolver-3 and AB-UPT is notably smaller than the gap AB-UPT originally showed over earlier methods, so the marginal benefit is more modest. Taken together, I agree with the authors that Transolver-3 may be preferable when one prioritizes accuracy on fixed-mesh scenarios, but practitioners should weigh these trade-offs. Personally, I find the physics attention approach of Transolver (1, ++, and 3) more elegant than the cross-attention mechanism of (AB-)UPT, so I am inclined to side with the authors.
> >
> > > Re Q6:
> >
> > Thank you for this additional data point! Could the authors comment on the impact of training on a fixed subset on the derived integral quantities (e.g., $C_d$, $C_l$)? This is also relevant in the context of my Q7 below.
> >
> > > Re Q7:
> >
> > Thank you for this clarification!
> >
> > I appreciate the analysis in Remark 3.1 and Figure 9, and I note that Figure 9(a) provides useful evidence that full-mesh state caching improves (full) field prediction accuracy (which partially justifies working on the full mesh). However, the argument in Remark 3.1 and Figure 9(b) specifically concerns quadrature error, and Eq. 6 bounds only this term. Since $C_d$​ and $C_l$​ are surface integrals, the quadrature error depends on the surface sampling density $N_s$​, not on volumetric resolution. If quadrature error is the dominant error source at lower evaluation resolutions (as Figure 9(b) suggests), would it not be more practical to predict on a denser set of surface points alone, rather than requiring full volumetric prediction at the original mesh resolution? I note that AB-UPT's neural field formulation would naturally support such surface-only refinement at inference time, whereas Transolver-3's mesh-bound architecture cannot easily densify the surface independently of the volume. I would appreciate the authors' perspective on whether surface-only refinement at evaluation time could be a viable and more efficient path forward for improving integral quantities.
> >
> > > Re Q9:
> >
> > Thank you for this additional experiment. Is this the default MGN (from Pfaff et al. 2021), or a more recent hierarchical variant with multi-scale coarsening? This distinction matters for interpreting the comparison, as hierarchical versions are considerably more expressive on large meshes.
> >
> > ---
> >
> > Overall, I remain positive about this work. I will consider raising my score after the authors commented on the follow-ups above.

---

> > > ### Author Response · Authors · 2026-04-04
> > >
> > > Dear Reviewer wQyX:
> > >
> > > Thank you very much for your response to our rebuttal. We are glad that our responses to Q1, Q3, Q4, Q5 and Q8 have addressed your concerns. Below are our responses to your follow-up questions.
> > >
> > > > Regarding **Q2:** AB-UPT offers the capability of being a conditional field, which Transolver-3 is not.
> > >
> > > Thank you for your insightful question. We clarify that Transolver-3 is also a native conditional field, capable of predicting physical quantities at arbitrary spatial coordinates. As detailed in $\underline{\text{Equation 5}}$, once the latent physical states are cached, the model can query any target coordinate $\mathbf{x}\in\mathbb{R}^{1\times C_{\text{in}}}$ to obtain the corresponding field value. This decoupled inference mechanism allows Transolver-3 to perform high-resolution surface-only refinement, providing the same continuous flexibility as the neural field formulation in AB-UPT.
> > >
> > > > Regarding **Q6:** impact of training on a fixed subset on the derived integral quantities.
> > >
> > > The mean absolute error (MAE) of $C_d$ and $C_l$ with full mesh training and fixed subset training is shown in the table below. The results demonstrate that training on a fixed subset results in higher prediction errors for $C_d$ and $C_l$, confirming that training with the full mesh is essential for accurately predicting integral quantities.
> > >
> > > | Quantities                   | $C_d$      | $C_l$      |
> > > | ---------------------------- | ---------- | ---------- |
> > > | Transolver-3 (single subset) | 0.0031     | 0.0072     |
> > > | Transolver-3 (full subset)   | **0.0025** | **0.0062** |
> > >
> > > > Regarding **Q7:** AB-UPT supports surface-only refinement at inference time, while Transolver-3's mesh-bound architecture cannot.
> > >
> > > Thank you for your constructive discussion. We clarify that Transolver-3 is not a mesh-bound architecture but rather a continuous conditional field, fully supporting surface-only refinement at inference time.
> > >
> > > As discussed in our response to **Q2**, Transolver-3’s decoupled inference allows for surface field prediction at arbitrary spatial coordinates, independent of the volumetric mesh resolution. We can fix the physical state cache and query an arbitrarily dense set of surface points. $\underline{\text{Figure 9 (b)}}$ specifically demonstrates this "surface-only refinement", where accuracy improves as the evaluation density increases while the latent cache remains constant.
> > >
> > > The inference latency of Transolver-3 also supports the practicality of surface-only refinement. Of the **260s** total inference time on DrivAerML, physical-state caching takes **~100s**, while field prediction takes **~160s**. Once the global cache is stored, retrieving or refining fields—whether for the entire volume or just a high-fidelity surface—becomes highly efficient.
> > >
> > > > Regarding **Q9:** About the experimental setting of MeshGraphNet.
> > >
> > > Thank you for your question. We use the default MeshGraphNet in the original response to Q9. We further evaluate the effectiveness of a hierarchical version of MeshGraphNet, and the results are summarized below.
> > >
> > > | Datasets                    | NASA-CRM |       | AhmedML |      |        |       | DrivAerML |      |        |       |
> > > | --------------------------- | -------- | ----- | ------- | ---- | ------ | ----- | --------- | ---- | ------ | ----- |
> > > | Variables                   | $p_s$    | $C_f$ | $p_s$   | $u$  | $\tau$ | $p_v$ | $p_s$     | $u$  | $\tau$ | $p_v$ |
> > > | MeshGraphNet (Default)      | 17.4     | 11.6  | 4.98    | 4.34 | 5.58   | 5.09  | 14.8| 16.3 | 21.4   | 23.2  |
> > > | MeshGraphNet (Hierarchical) | 14.8     | 9.81  | 3.72    | 3.97 | 5.01   | 4.50  | 14.3      | 15.0 | 20.1   | 21.8  |
> > > | Transolver-3 | **8.71** | **5.85** | **2.96** | **1.60** | **4.81** | **2.16** | **3.71** | **4.14** | **5.85** | **5.72** |
> > >
> > > The results show that a hierarchical structure surpasses the default version, while Transolver-3 still consistently outperforms this hierarchical version across all datasets.
> > >
> > > Thank you again for your time and dedication.

---

### Decision · Program_Chairs · 2026-04-30

**Decision:**

Accept (regular)

**Comment:**

After discussion and careful consideration of the rebuttal, the paper has been accepted. Congratulations.

The reviewers generally agree that the paper presents a highly impactful and practically valuable solution. The rebuttal thoroughly addressed major concerns regarding statistical significance, baseline comparisons, ablation studies, and generalization to discontinuous and shock‑dominated flows, which significantly strengthened confidence in the work.

All revisions in the rebuttal should be incorporated into the revision, including:

- Key experimental results, including ablation studies and generalization to complex physical fields, should be strengthened to better highlight the unique advantages of the proposed method.
- A limitations section should be added to discuss tradeoffs, boundary conditions, and open challenges clearly.
- The description of efficiency and practical gains, especially wall‑clock time, memory usage, and scaling behavior against strong baselines such as AB‑UPT, should be clarified and made more concrete.
- The sampling strategy in amortized training and numerical stability under large‑scale chunk caching should be explicitly clarified.